# Drivers of female power in bonobos
Martin Surbeck [1,2] ✉, Leveda Cheng[1], Melodie Kreyer[3], Gerrit Gort[4], Roger Mundry[5,6,7], Gottfried Hohmann[2,3] & Barbara Fruth [3,8]

In mammals, female dominance over males is a rare phenomenon. However, recent findings indicate that even in species with sexual dimorphism biased towards males, females sometimes occupy high status. Here we test three main hypotheses explaining intersexual power relationships, namely the self-reinforcing effects of winning and losing conflicts, the strength of mate competition, and female coalition formation. We test these for bonobos (*Pan paniscus*), one of our closest living relatives, where females have high status relative to males despite male-biased size dimorphism. We compiled demographic and behavioral data of 30 years and 6 wild living communities. Our results only support predictions of the female coalition hypothesis. We found that females target males in 85% of their coalitions and that females occupy higher ranks compared to males when they form more frequent coalitions. This result indicates that female coalition formation is a behavioral tool for females to gain power over males.

Males are dominant to females in most social mammals[1]. Typically, dominance ranks are inferred by observers based on subordinance signaling and/or the outcome of aggressive interaction on a dyadic level[2,3]. Based on the concept of power established by Simon[4], Lewis[5] identified factors that influence an individual's likelihood to win dyadic interactions and consequently affect its status. These factors include an individual's fighting ability (referred to as dominance), which depends on the ability to use physical force and the likelihood of getting agonistic support[6]. In addition, they include aspects that influence leverage, such as control over mating opportunities and possession of special knowledge[5,7,8]. Furthermore, the history of previous interactions can be relevant to the outcome of dyadic interactions e.g. [9]. Due to differences in the nature of the competition among males and among females, the sexes are often placed in different hierarchies, preventing the quantification of the power relations between them. However, intersexual power can have consequences for the use of coercive mating strategies and infanticide by males, female choice, and the access to resources[10,11]. Identifying proximate mechanisms underlying variation in power between individuals of both sexes may help to identify factors favoring the emergence of power of one sex over the other.

Many mammal species, including the majority of primate species show male-biased sexual dimorphism in body size and fighting ability[1,12]. Greater contest competition among males than among females presumably accounts for this difference[10]. The evolutionary mechanisms leading to female power over all males within a group, a rare phenomenon among social mammals, seem species specific and sometimes associated with an

absence of male-biased sexual dimorphism such as in spotted hyenas (*Crocuta crocuta*) and lemur species (*Lemoroidea spp*)[7,13,14]. However, power relationships between the sexes vary between groups in several species despite male-biased sexual dimorphism, including vervet monkeys (*Chlorocebus pygerythrus*)[11,15], macaques (*Macaca spp*)[16], capuchin monkeys (*Sapajus libidinosus; S. nigritus, and S. xanthosternos*)[17] and rock hyraxes (*Procavia capensis*)[18]. This variation allows for testing hypotheses on the mechanisms that influence the degree to which females have more power than males[19].

Three main hypotheses have been put forward to explain the variation in female power over males within groups: The self-organization hypothesis, the reproductive control hypothesis, and the female coalition hypothesis.

The **self-organization hypothesis** is built on the self-reinforcing effect of winning and losing conflicts. Self-reinforcement of conflict outcomes implies that individuals are more likely to lose after having lost a conflict and more likely to win after having won a conflict[20–22]. According to the self-organization-hypothesis based on sex ratio, females can be dominant over several males, despite their smaller body size than males. High female power over males has been shown particularly in societies with a large proportion of males. The high rates of male-male conflicts result in some males losing competitive strength over the course of frequently lost conflicts and dropping below certain females in their winning ability and their position within the hierarchy[11,16]. The association between the degree of female power over males and proportion of males within a group is shown in DomWorld, an agent based model of this process[23], and has been empirically supported by

[1]Harvard University, Department of Human Evolutionary Biology, Cambridge, USA. [2]Max Planck Institute for Evolutionary Anthropology, Leipzig, Germany. [3]Max Planck Institute of Animal Behavior, Konstanz, Germany. [4]Biometris, Wageningen University and Research, Wageningen, Netherlands. [5]Cognitive Ethology Laboratory, German Primate Center, Leibniz Institute for Primate Research, Göttingen, Germany. [6]Department for Primate Cognition, Georg-August-University Göttingen, Göttingen, Germany. [7]Leibniz-Science Campus Primate Cognition, Göttingen, Germany. [8]Centre for Research and Conservation, Royal Zoological Society of Antwerp, Antwerp, Belgium. ✉e-mail: msurbeck@fas.harvard.edu

studies in several species[11,15–18], including humans where gender composition of groups affected the relative influence of women versus men on decision making[24]. An alternative explanation to the link between sex-ratio and female power within a group might be that a decreasing female biased sex-ratio changes the supply and demand for mating opportunities and reduces the leverage of females[25].

The **reproductive control hypothesis**, concerns mechanisms favoring non-aggressive male mating strategies[10,26]. According to the hypothesis, males are incentivized to rely on non-aggressive strategies, if their ability to monopolize potentially fertile females is reduced. This reduction in the ability to monopolize potentially fertile females can be the result of several eco-evolutionary pathways, including concealed ovulation or synchronization of female reproduction[10]. Hence, as a consequence of increased female leverage over reproduction, male aggression against females in the context of mating may no longer be adaptive[5,7,25,27], and female position in the power hierarchy rose relative to males[28]. Partial empirical support for a link between control over mating opportunities and intersexual power relations comes from studies on Verreaux's sifaka (*Propithecus verreauxi*)[25] and Vervet monkeys (*Chlorocebus pygerythrus*)[29].

The **female coalition hypothesis** implies the formation of female alliances and a derived female power based on coalitionary aggression against males[5]. While this hypothesis is the least discussed, the formation of female coalitions against males may reduce sex differences in (agonistic) power in some cases, resulting in females winning a higher proportion of dyadic conflicts against males and consequently rising in rank[30–34]. Empirical support for the link between the number of allies and a high female dominance over males comes from spotted hyenas, where females usually remain in their natal communities[35].

The goal of the present study was to investigate these hypotheses in bonobos (*Pan paniscus*) to understand which mechanisms underly the observed high degree of female power over males. Bonobos are one of our closest living relatives, they live in multi-male multi-female groups (also referred to as communities), have a male-biased sexual dimorphism and differ from all other great ape species in that females occupy high ranks within their communities[27]. In bonobos, even though females are usually the migrating sex, they can have priority of access to food resources[34], they can outrank adult males, and the highest ranks among adults are occupied by females[27,36–38]. This dynamic between the sexes is in striking contrast to chimpanzees (*Pan troglodytes*), their sister species, where males dominate all females before they are fully mature[39,40]. Although the high degree of female power over males may be a derived trait[41], in bonobos, the actual underlying mechanisms are still debated.

Firstly, the self-organization hypothesis with its self-reinforcing process of winning and losing conflicts has been proposed to underly the differences in power dynamics between the sexes in bonobos and chimpanzees[42]: The more cohesive bonobo communities and the higher number of competitive interactions among males in the DomWorld model resulted in females ranking higher compared to the less cohesive chimpanzee communities. Furthermore, the bonobo system seems to adhere to many underlying assumptions of the model, including a greater fighting capacity of males than females, higher rates of aggression in males and high variation in the percentage of males. Here we test a different prediction derived from DomWorld that the degree of female power over males increases with a higher proportion of males in the group[11,16–18] and we predicted a higher degree of female power over males in communities with a higher proportion of males.

Secondly, the reproductive control hypothesis in bonobos has been proposed based on the patterns of sexual signaling in females. The long and frequent periods of maximal tumescence of female sexual swellings provide an unreliable indicator of ovulation compared to chimpanzees and other primates[43]. The extended time periods of potential receptivity[44] and the extensive overlap in female sexual receptivity[28] lead to high costs of male mate guarding and, according to the priority of access model[45], reduces the potential of single males to monopolize access to potentially fertile females. Consequently, non-aggressive male mating strategies maximizing time

spent around maximally tumescent females, may increase females' tolerance towards these males and results in more mating opportunities and possibly higher reproductive success for males. In support of a link between mate competition and the degree of female power over males, a previous study showed that a bonobo female is more likely to win a dyadic conflict against a male when she exhibits a maximally tumescent swelling[27]. However, whether a given male's potential to monopolize females contributes to the variation in the degree of female power over males within and across bonobo communities remains unclear. According to the reproductive control hypothesis we expected a higher degree of female power over males within and across communities when multiple maximal tumescent females are present at the same time, because females become less monopolizable by individual males.

Thirdly, the female coalition hypothesis in bonobos is based on several studies that indicate the relevance of female coalitions underlying the high degree of female power over males[30,34]. Bonobo females have been described to form agonistic coalitions against males that can result in injuries of males[46,47]. These coalitions sometimes are triggered by male aggression against females or their infants, but seem not equally prominent across different study communities[27,46,48]. According to this hypothesis, we predicted a higher degree of female power over males to be associated with a higher frequency of females to engage in coalitionary aggression, particularly against males.

In the present study we make use of a dataset from six bonobo communities from three field sites, with observation periods ranging between two and six years for a given community. We test the three hypotheses investigating the potential influence of the outlined mechanisms on the variation in the degree of female power over males within and between communities. We use the full set of data available from all six communities (30-years-6-communities dataset) to test the prediction of the **self-organization hypothesis** regarding the influence of the percentage of males in a community. We use a smaller and more detailed dataset available from five of the six communities (15-years-5-communities dataset) to test in addition the prediction of the **reproductive control hypothesis** regarding the influence of the number of maximally tumescent females, and the prediction of the **female coalition hypothesis** regarding the frequency of female coalitionary aggression. In our analysis we found only support for the female coalition hypothesis. Specifically, the propensity of females to form coalitions against males was positively associated with the degree of power that females had over males and consequentially our results highlight the relevance of agonistic support among bonobo females for their power over males.

## Results

Bonobo females often had power over males, although not exclusively. Quantifying the degree of power of females over males by the proportion of intersexual conflicts in which males submitted to females: in the 30-years-6-communities dataset, male bonobos submitted to females on average in 61% of these conflicts ($N = 1786$ conflicts; SI Table 1 in the supplementary information). The percentage of intersexual conflicts in which males submitted varied considerably within and across communities with maxima of 100% in the Eyengo community in Lomako during 1998 and 98.4% in the Kokoalongo community in Kokolopori during 2020, and minima of 18.2% in the Ekalakala community in Kokolopori in 2016 and 21.4% in the Bompusa East community in LuiKotale in 2014 (SI Table 1). Intersexual conflicts occurred in each community in multiple dyads and the average number of intersexual dyads with conflicts within a given community was on average 16 (Range = 4–59; SI Table 1).

Quantifying the degree of female power over males by the proportion of males in a community outranked by each female in the 15-years-5-communities dataset (FDI DS), values varied considerably within and across those five communities and were 0.70 on average (Range 0.33–1). The maximum was found in the Kokoalongo community during 2020, with all adult females ranking higher than all the males, and the minimum in the Ekalakala community during 2018, with an average female outranking

about one third of the males (SI Table 1). When we calculated the relationships with DS within a community in a given year, in 46.7% of community years, a female occupied the highest rank. Both measures of female power, the proportion of conflicts won by females and the FDI-DS, correlate strongly in our 15-years-5-communities dataset (Spearman's rank correlation rho = 0.89, P < 0.001; SI Fig. 1).

## Association between the proportion of males and the degree of female power over males within communities

The proportion of males varied considerably within and across communities with a minimum of 0.24 and a maximum of 0.57 (Fig. 1; SI Table 1). A comparison of the full model and a reduced model comprising all the variables except the predictor variable of proportion of males within communities revealed no evidence for a relationship between the percentage of the intersexual conflicts in which males submitted to females within a community/year and the proportion of males/year within the community ($\chi 2 = 0.000$, df = 1, P = 0.997; Table 1), indicating that the proportion of males was not associated with the variation in the degree of female power over males in the large 30-years-6-communities dataset (Fig. 1). Further, our data showed no evidence that the proportion of subadult males in a

## Table 1 | The table shows an overview of structure and result of the model analyzing differences in degree of female power over males within communities in relation to the proportion of males

| Term | Estimate | SE | CL$_{lower}$ | CL$_{upper}$ | $\chi^2$ | df | P |
|---|---|---|---|---|---|---|---|
| (Intercept) | 0.514 | 0.303 | −0.097 | 1.108 | | | |
| proportion males[a] | 0.001 | 0.229 | −0.425 | 0.395 | 0.000 | 1 | 0.997 |

Sample size consists of 30 combinations of community and year; 6 communities; 1099 conflicts won by females and 687 conflicts won by males. Dispersion parameter: 0.439.
[a]All covariates were z-transformed to a mean of zero and a standard deviation (sd) of one; mean (sd) of the original variables for proportion males: 0.354 (0.086).

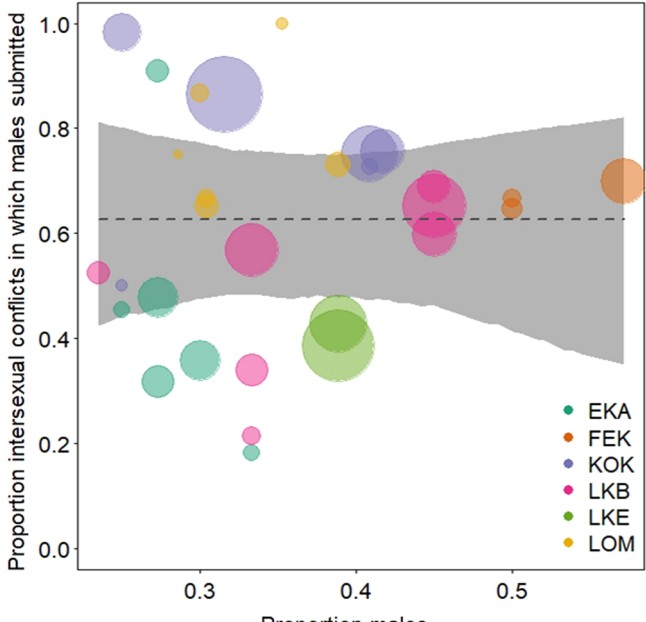

**Fig. 1 | The figure shows the relationship between the proportion of males within a community/year and the percentage of all intersexual conflicts in which females received submission from males.** Colors depict communities Ekalakala (EKA), Fekako (FEK), Kokoalongo (KOK), Bompusa West (LKB), Bompusa East (LKE), Eyengo (LOM) and circle size represents number of conflicts.

community (≥10 years and ≤15year) influenced the percentage of the intersexual conflicts in which males submitted within a community year ($\chi 2 = 0.066$, df = 1, P = 0.796). Finally, the proportion of males was also not associated with variation in both measures of female degree of power over males in the 15-years-6-communities dataset (2 C: $\chi 2 = 0.013$, df = 2, P = 0.910; 3 C: $\chi 2 = 0.122$, df = 2, P = 0.727; SI Table 2).

## Female swelling patterns and the degree of female power over males within communities

The synchrony of female sexual swellings quantified by the presence of multiple females exhibiting maximal tumescence varied substantially within and between sites with more than one maximally tumescent female present in a given community on average 40% of the yearly observation days (Range: 5–73% of the days; SI Table 1).

Both models, which used the percentage of intersexual conflicts where females received submission from males in a community per year as the response variable (Models 2 A/B), demonstrated a significant difference between the full and null models (Model 2 A: $\chi^2 = 15.561$, df = 2, P < 0.001; Model 2B: $\chi^2 = 12.350$, df = 2, P = 0.002; Table 2). However, both models revealed no evidence that the percentage of days with more than 2 maximally tumescent females (2MTF) had an influence on the degree of female power over males (Model 2 A: Estimate = −0.032, SE = 0.366, p = 0.930; Model 2B: Estimate = −0.051, SE = 0.197, p = 0.796; Table 2). The models with the response variable representing the average proportion of males in a community outranked by each female (Models 3 A/B) showed no significant difference between the full and null models (model with a random slope for 2MTF [Model 3 A: $\chi^2 = 4.603$, df = 2, P = 0.100]; model with a random slope for the frequency of all female coalitions [FFC; Model 3B: $\chi^2 = 3.016$, df = 2, P = 0.221]; Table 2). Further, both models revealed no evidence for an influence of the percentage of days with more than 2 maximally tumescent females (3 A: Estimate = −0.073, SE = 0.370, p = 0.819; 3B: Estimate = −0.055, SE = 0.246, p = 0.816; Table 2).

## Female coalition formation and the degree of female power over males within communities

The occurrence of female coalitions varied substantially within and across sites. The yearly average rates of female coalitionary aggression were one per 203.5 observation days in the Fekako community in Kokolopori (Range: none within a year - one every 156 observation days), one per 125.1 days in the Ekalakala community in Kokolopori (Range: every 10.9–346.0 days), one per 58.4 days in the Bompusa East community in LuiKotale (Range: every 42.8–74 days, one per 22.1 days in the Bompusa West community in LuiKotale (Range: every 14.7–32 days), one per 11.1 days in Kokoalongo community in Kokolopori (Range: every 1.9–22days, SI Table 1). In 85% of the cases, female coalitions targeted males (Kokolopori: 78% in Kokoalongo, 90% in Ekalakala, 100% in Fekako; LuiKotale: 91% in Bompusa West and 100% in Bompusa East).

Both models considering the percentage of conflicts in which males submitted to females (2 A/B) showed similar results concerning the fixed effect of female coalition (Table 2): the model with 2MTF as random slope revealed a significant positive influence of the frequency of all female coalitions (FFC) on the degree of female power over males (Fig. 2a; Model 2 A: Estimate = 0.762, SE = 0.146, p < 0.001; Table 2) and the model with FFC as random slope revealed a positive trend for the influence of the frequency of female coalitions (Model 2B: Estimate= 0.723, SE = 0.318, p = 0.059).

For the models with the response variable indicating the average proportion of males in a community outranked by each female (Models 3 A/B), similar results were obtained, showing a positive effect size of the frequency of female coalitions on the degree of female power over males (Table 2/Fig. 2b; Model 3 A: Estimate = 0.460, SE = 0.415, p = 0.036; Model 3B: Estimate = 0.430, SE = 0.220, p = 0.083).

Restricting the analysis to the frequency of female coalitionary aggression that targeted males (FFCm), rather than FFC, revealed very similar results. The models with random slope of 2MTF revealed a significant difference between the full- and the null-models (Model 2A$_{ff \to m}$:

**Table 2 | A/B shows an overview of the structure and results of the 4 models analyzing the link between the two measures of the degree of female power over males (Models 2: Intersexual conflicts in which males submitted to females within a community/year; Models 3: Average percentage of males in a community outranked by each female [based on David Score]) and the predictor variables of the propensity of females to form coalitions and the synchrony of maximally tumescent females**

| A | Model 2 A | | | | | Model 2B | | | | |
|---|---|---|---|---|---|---|---|---|---|---|
| | 5 communities, total 15 years | | | | | | | | | |
| Response | Intersexual conflicts in which males submitted to females within a community/year | | | | | Intersexual conflicts in which males submitted to females within a community/year | | | | |
| Full-null model | LRT, χ2 = 15.561, df = 2, P < 0.001 | | | | | LRT, χ2 = 12.350, df = 2, P = 0.002 | | | | |
| | Est | SE | CI$_{low}$ | CI$_{high}$ | P | Est | SE | CI$_{low}$ | CI$_{high}$ | P |
| Intercept | 0.803 | 0.199 | 0.422 | 1.153 | | 0.48 | 0.187 | 0.092 | 0.886 | |
| Female coalition frequency | 0.762 | 0.146 | 0.488 | 1.097 | <0.001 | 0.723 | 0.318 | 0.126 | 1.389 | 0.059 |
| Percentage days with 2 or more maximally tumescent females | −0.032 | 0.366 | −0.743 | 0.676 | 0.93 | −0.051 | 0.197 | −0.495 | 0.367 | 0.796 |
| Random factor | Community, observation level | | | | | Communities, observation level | | | | |
| Random slope | Percentage days with 2 or more maximally tumescent females | | | | | Female coalition frequency | | | | |
| B | Model 3 A | | | | | Model 3B | | | | |
| | 5 communities, total 15 years | | | | | | | | | |
| Response | Average proportion of males in a community outranked by each female [based on David Score] | | | | | Average proportion of males in a community outranked by each female [based on David Score] | | | | |
| Full-null model | LRT, χ2 = 4.603, df = 2, P = 0.100 | | | | | LRT, χ2 = 3.016, df = 2, P = 0.221 | | | | |
| | Est | SE | CI$_{low}$ | CI$_{high}$ | P | Est | SE | CI$_{low}$ | CI$_{high}$ | P |
| Intercept | 1.09 | 0.268 | 0.678 | 1.537 | | 0.995 | 0.28 | 0.558 | 1.518 | |
| Female coalition frequency | 0.46 | 0.415 | 0.092 | 0.861 | 0.036 | 0.43 | 0.22 | −0.019 | 0.921 | 0.083 |
| Percentage days with 2 or more maximally tumescent females | −0.073 | 0.37 | −0.624 | 0.51 | 0.819 | −0.055 | 0.246 | −0.54 | 0.437 | 0.816 |
| Random factor | Community, observation level | | | | | Community, observation level | | | | |
| Random slope | Percentage days with 2 or more maximally tumescent females | | | | | Female coalition frequency | | | | |

Models A and B differ in the structure of the random slope term.

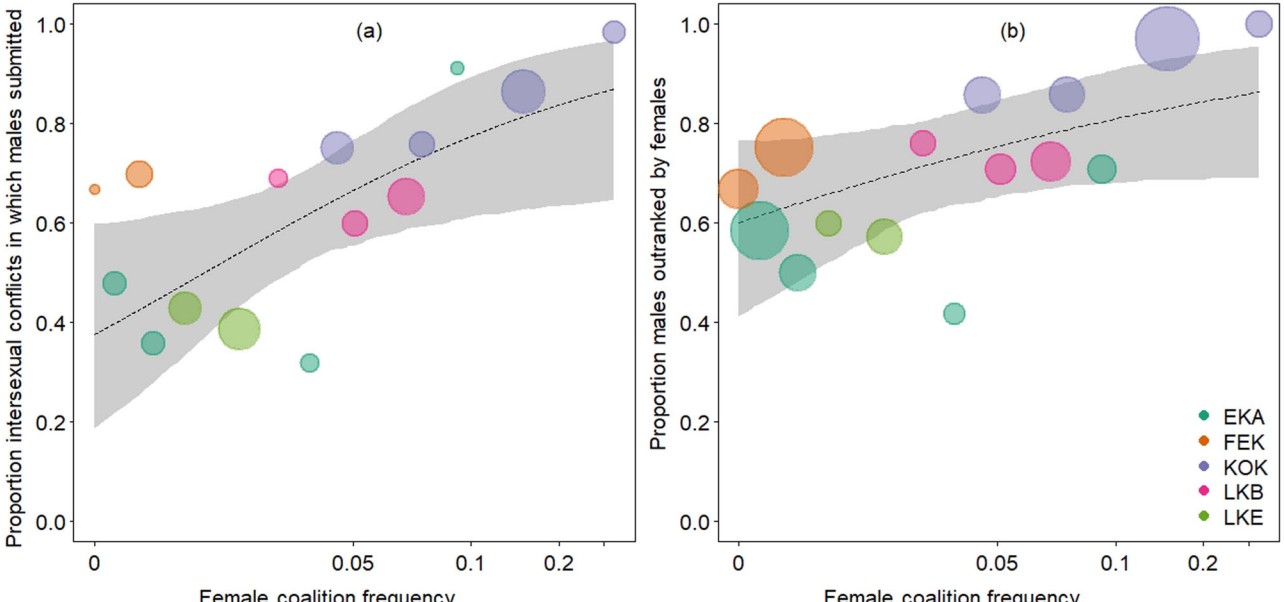

**Fig. 2 | The figure shows the relationship between two different indices of degree of female power over males and the frequency of female coalition formation. a** shows the relationship between frequency of all female coalition formation and the proportion of conflicts in which females received submission from males submitted to females; and **b** shows the relationship between frequency of all female coalition formation and the average proportion of males in a community outranked by each female. The dashed line and the grey polygon show the fitted model and its confidence limits for the percentage of days with more than two maximally tumescent females being at its average (model estimates from the models with random slope of FFC). Area of dots is proportionate to number of agonistic interactions observed between sexes (**a**; range 15–235) and the overall number of conflicts with submissions (**b**; range 58–493) during a given year. Colors depict the communities Ekalakala (EKA), Fekako (FEK), Kokoalongo (KOK), Bompusa West (LKB), Bompusa East (LKE).

$\chi 2 = 16.395$, $df = 2$, $P < 0.001$; Model $3A_{ff \to m}$: $\chi 2 = 9.483$, $df = 2$, $P = 0.009$; SI Table 3A/B) and a significant positive association between the frequency of female coalitionary aggression against males and both measures of the degree of female power over males (Model $2A_{ff \to m}$: Estimate= 0.784, SE = 0.152, $p < 0.001$; Model $3A_{ff \to m}$: Estimate= 0.491, SE = 0.124, $P = 0.001$; SI Table 3A/B). While the full– and the null- model for the models with a random slope of FFCm were not significantly different (Model $2B_{ff \to m}$: $\chi 2 = 3.170$, $df = 2$, $P = 0.205$; Model $3B_{ff \to m}$: $\chi 2 = 4.753$, $df = 2$, $P = 0.092$; SI Table 3A/B), both models revealed a positive effect sizes of the frequency of female coalitions against males and both measures of the degree of female power over males (Model $2B_{ff \to m}$: Estimate= 0.819, SE = 0.209, $p < 0.001$; Model $3B_{ff \to m}$: Estimate= 0.449, SE = 0.171, $p = 0.009$; SI Table 3A/B).

## Discussion

Here, we compiled demographic and behavioral data of 6 wild living bonobo communities spanning 30 years of observation, to investigate mechanisms underlying female power over males, a rare phenomenon in mammalian societies. From the three tested hypotheses, we found only support for the female coalition hypothesis. Specifically, we found that the propensity of females to form coalitions against males varied considerably, both within communities over time and across communities, and that this variation was positively associated with the degree of power that females had over males: Communities and years in which adult females formed more coalitions corresponded to females winning more dyadic conflicts against males and ranking higher than males, supporting the female coalition hypothesis. In contrast, we neither found support for the self-organization-hypothesis nor for the reproductive control hypothesis because the percentage of males within communities and the synchrony of female sexual swellings did not correlate with our measures of degree of female power over males. Taken together, our results highlight the relevance of agonistic support as a behavioral strategy for females to gain power over males.

### Female power relative to males

Our study shows that an average bonobo female in the wild won 61% of the conflicts with subadult and adult males and outranked about 70% of those males. However, both indices of female power over males varied largely over communities and times. The published percentage of 38% decided agonistic intersexual conflicts won by females from Wamba (inferred from AS and AAS in Fig. 2)[49], a long term research site not included in our study, falls within the range documented here. In all our study communities, a male had the highest rank at least for some years, which differs from some previous studies from captivity, where females were consistently reported to occupy the highest ranks[36–38]. Several factors could explain the discrepancy between these findings from captive studies and our results. Firstly, the demographic composition of social communities differs between captivity and the wild. In the wild, males more consistently remain in their natal communities, offering them the potential for long-term allies or connections, especially with their mothers[49,50]. In three communities—Ekalakala, Bompusa West, and possibly Bompusa East—the mothers of the highest-ranking males were present. In contrast, in the remaining two communities, Kokoalongo and Fekako, none of the adult males had a living co-residing mother. Secondly, competition over food might occur more frequently in captivity and bonobo females have been shown to exhibit greater power in a feeding context[28]. Finally, the fission-fusion dynamics in the wild provide better opportunities to avoid severe conflicts, and males sometimes leave the community after being attacked by females for several days or even weeks (MS personal observation).

Overall, despite the variation in the degree of female power over males both within and across communities, the implications of this variation for individual bonobos and their social dynamics remain poorly understood. Potential implications include differentiated access to resources[34], the effectiveness of sexual coercion as a male mating strategy[10] and influence on reproductive success of offspring[50]. Some studies suggest that females may have priority access to food regardless of their individual rank[34], while others

find a positive association between access to monopolizable food resources and intersexual rank[27].

While male chimpanzees increase their power over females and employ coercive mating strategies during adolescence[40], studied bonobo populations show no evidence of males using coercive mating strategies at any age. Instead male aggression against maximally tumescent females is rare[27]. One implication of higher ranks of individual females relative to males may be that it affects the dynamics of mate competition through more efficient support of their mature sons[49–51].

### Self-organization-hypothesis with sex ratio

We did not find a link between the proportion of males within communities and the degree of female power over males that is predicted by simulation outcomes in DomWorld, which relies on the self-reinforcing effects of winning and losing fights[16]. This lack of correlation contrasts with findings in capuchins, vervet monkeys, macaques and hyraxes[11,15–18]. In line with further predictions of the model, studies in vervet monkey and capuchin monkey confirmed that a higher proportion of males within groups is not only linked to increased conflict rates among males, but also a greater proportion of female victories over males compared to all other adult pairings[11]. Female rock hyrax occupy alpha position exclusively in more than half of the groups and female power over males increased with the percentage of males in the subset of groups including multiple males[18]. Various explanations can account for the lack of a relationship between the proportion of males in a community and the measures of female power in bonobos. Firstly, the fission-fusion dynamic potentially leads to different sex-ratios in parties than in the community as a whole[52]. However, we found that the percentage of males within communities correlated well with the percentage of males within parties overall (see SI Fig. 2)[28]. Secondly, the variation in community composition potentially prevents the emergence of a higher degree of female power over males like in the model DomWorld (35). Specifically, the observed fission-fusion dynamic corresponds to a reduced group density in DomWorld, which in the virtual environment results in fewer interactions among specific group members, a weaker hierarchy and lower degree of female power over males[42]. Note that such a fission-fusion dynamic is absent in capuchins, vervet monkeys and macaques. Finally, a key distinction between the previously studied species that align with the predictions of the self-organization hypothesis and bonobos could be that the influence of female coalitions against males overshadows the significance of the male-to-female ratio. Bonobo females seem to have developed a behavioural strategy allowing them to influence their ranking relative to males within communities.

### Synchrony of female sexual swellings

In a previous study in the Bompusa West community, LuiKotale, we found that changes in the degree of tumescence of sexual swellings affected a female's likelihood to elicit submissive behaviour during conflict with a male[27]. The periods of tumescence in sexual swellings in female bonobos are an unreliable signal of ovulation, with ovulation falling outside the maximal tumescent periods in one third of the cases[43]. We therefore assumed that the link between a reduction of the predictability of ovulation and the associated reduced potential of males to monopolize a given female underlies changes in the leverage that females have over males. In addition to the unpredictable timing of ovulation, bonobo females are particular in that they exhibit visual signals of fertility over prolonged periods, promoting an increased overlap between females in their ovulation signal[28], which also increase the potential costs of mate guarding to males, decrease their potential to monopolize access to a female and reduces the relevance of male aggression. However, the findings from our study do not indicate that intercommunity differences in this signaling overlap between females influence our overall measures of the degree of female power over males. We used the previously established measurement of percentage of days with more than two maximally tumescent females in a community to quantify the female monopolization potential by males and the potential relevance of male aggression. While this study is a good initial step, future investigations trying to quantify

community differences in the relevance of male aggression should include additional measures, such as reliability of sexual swellings or the responsiveness of male androgen levels to the presence of maximally tumescent females.

In other species, such as the Verreaux's sifaka, females gain power over males not only through an increase in leverage when becoming sexually mature and during the mating season, but also when the sex ratio in groups was male-biased[25,53]. In bonobos, factors such as year-round reproduction, confused ovulation, and promiscuous mating may change the dynamics of mate competition, and the sex ratio or even the ratio of males to maximally tumescent females becomes less indicative of male mating opportunities. However, an overall reduction in male investments in aggressive mate competition because of changes in female sexual signaling remains a viable scenario for the high degree of female power over males in bonobos on an evolutionary scale[28]. This idea is supported by findings of a relatively reduced sexual dimorphism in body and canine sizes[54–59] and a lack of an increase in testosterone during mate competition[60,61] in bonobos compared to chimpanzees, a species for which females sexual swellings are more reliable indicators of fertility[43,62,63].

The observed association between female reproductive autonomy and increased female power in bonobos is in line with evidence across multiple mammal species in which females have high status[10]. However, the reproductive autonomy, resulting from a decrease in the potential of individual males to monopolize reproductive access to given females against their will, is achieved in different ways in different species. For examples in fossas (*Cryptoprocta ferox*), females choose mating places high in trees that limit access to males and give them competitive advantages them[64]. In spotted hyenas (*Crocuta crocuta*), the peculiar morphology of the female genitals requires full cooperation from females during copulations[65].In several lemur species very short reproductive periods make it energetically affordable for females to resist male mating attempts[66] and a larger body size may contributes to females being able to reject unwanted male advances[14, but see 16]. Finally, we note that rarely does one sex has complete autonomy over reproduction and it is therefore not surprising to observe rare cases of forced copulations[67] and infanticide, even in lemurs species with high female power[68].

### Female coalitionary aggression
Initially, female coalitionary aggression in our study included all coalitions that formed between females regardless of the sex of the victim. We chose this first measurement because it best characterizes the general propensity of bonobo females to form coalitions in the first place. In 85% of the cases, males were the victims of female coalitions. We can discount the possibility that including also female coalitions against females biased our findings, because restricting our analysis to only female coalitions against male victims did not alter the results. In all populations we sometimes observed males participating in female coalitions. The participating males never were in leading positions but rather trailed behind several female aggressors (MS personal observation). The male participation appears highly unlikely to have impacted any of the observed dynamics or consequently affected our findings. Female coalitions may be used to direct intense contact aggression at males sometimes causing severe injuries[46,48]. Consequently, males at times avoid associating with members of the community for weeks (MS personal observation). Potentially 2 out of 57 observed female coalitionary aggressions against males between 2016 and 2018 in Ekalakala and Kokoalongo resulted in male injuries (one observed and one suspected), but variation across years and communities in the rates of wounding might occur.

Overall, the propensity of females to form coalitions varied substantially among communities and to some degree within communities. At Wamba, 58 acts of female coalitionary aggression have been observed over the course of four observation periods lasting for about half a year each. Therefore, the frequency of female coalitions in this population not included in our study falls likely within the range captured here[46]. The documented between-community variation is unlikely to be explained by differences in the observation protocol, because data collection at Kokolopori is based on

the one at LuiKotale and in the case of Kokolopori, observers rotated between the different communities. Furthermore, the derived numerical differences in female coalition formation matched observer intuition about the propensity of females in the respective communities to attack males in the respective years. The generally lower variation within communities may arise from the fact that the study duration in none of the communities exceeds four years and therefore temporal variation arising from demographic change does not manifest.

Triggers of female coalition formation seems to vary between sites, with female coalitions mostly formed after male aggression against mature females in Wamba[46], and after male aggression against offspring in the Eyengo, Lomako[48], and the Bompusa West, LuiKotale[27], communities. In many cases an actual trigger seems very hard to identify[27]. A hypothesized function of the female coalitions is that they select against the expression of male aggression[69,70]. While direct comparison of aggression rates of bonobos and chimpanzees suggest partial overlap in the overall frequency of certain types of male aggression[71–73], future studies on the context of coalition formation would be helpful to identify their proximate functions and their influence on the expression of intense forms of male aggression.

While previous studies in bonobos did not find evidence for a link between individual female ranks relative to males and the availability of specific coalition partners[27], we find in this study that aspects of alliance formation, such as composition and frequency, modify intersexual power relationships. Strong females bonds and individual variation in the ability to recruit social allies have been shown to underly the observed variation in ranks between the sexes in spotted hyenas, where females largely remain in their natal community[35,74,75]. Specifically, spotted hyenas with higher potential social support won dyadic conflicts, independent of body mass or sex. The observed female dominance was thus attributed to an imbalance in social support that favored females, stemming directly from the tendency of males to disperse and the subsequent disruption of social bonds[35]. Interestingly, our results now indicate that female coalitions can serve as behavioral tool to increase and/or maintain a high degree of female power over males even in the absence of female philopatry. Understanding the factors underlying the variation in impact of female coalition formation on intersexual dominance relations, as for example the observed differences between bonobos and chimpanzees[76], remains for future studies.

### Conclusion
Despite a long-standing interest in species with apparent high degree of female power over males, the understanding of the drivers and actual relevance of intersexual power is still in its infancy. In this study of bonobos, a species with male-biased sexual dimorphism, male philopatry, and a fission-fusion social organization, we show that female propensity to form coalitions can account for the variation in the degree of female power over males across time and communities. This finding underscores the importance of mutual social support among unrelated females in shaping intersexual dominance dynamics in one of our closest living relatives. It adds to the diversity of evolutionary mechanisms associated with an increase in female power. Some established hypotheses to explain female power suggest the self-organizing principle of the winner-loser effect in group-living species[21,23], while others emphasize mechanisms that empower females to use fertilizable eggs as leverage in interactions with males[5]. Pathways enhancing female leverage can include physiological traits like a larger body size relative to males[14], unique genital morphology requiring female mating cooperation[9], and a reproductive physiology concealing ovulation[43,77]. They furthermore included behavioral factors such as the timing and location of reproduction that prevent male monopolization of mating opportunities and reduce the relevance of male aggression[64,66,78,79]. Finally, evidence from spotted hyenas suggests that differences in social support between the sexes enhances power in the philopatric females[35]. Generally, many of these mechanisms seem not entirely independent. For instance, an initially low sexual dimorphism can foster alternative pathways to female empowerment, such as through the formation of coalitions[35]. Future research should explore the interplay of various sources of female power.

## Methods

### Study sites and subjects

We compiled demographic and behavioral data from six bonobo communities in three populations: the Eyengo community (1993-1997, 1998) at Lomako, the Bompusa West community (former Bompusa; 2007-2010, 2013-2015), and the Bompusa East community (former East; 2018-2019) at LuiKotale, and the Ekalakala community (2016-2021), the Kokoalongo community (2016-2021) and the Fekako community (2019-2021) at Kokolopori. Behavioral data consists of all observed acts of aggression. We included all subadult and adult individuals estimated to be 10 and older in this study, which corresponds to the youngest male age known to reproduce in the wild[80]. While males at this age might not be fully grown, they already independently interact with the adult group members and the earliest known age of an alpha male is 13 years at Wamba[81]. Most females at this age likely already dispersed from their natal community[82]. Community composition is shown in SI Table 1. The median community size of individuals (≥10 years) in our sample was 17 (Range: 6–26), consisting of a median of 6 males (Range: 3–10) and 11 females (Range: 3–16; SI Table 1). The total dataset with the above information consists of 30 observation years across six communities (30-years-6-communities dataset). For all but the Eyengo community, we have observation years that include data on female swelling scores, as well as all-occurrence observations of female coalitions, which make up a reduced dataset of 15 observation years across five communities (15-years-5-communities dataset; SI Table 1).

Daily subgroups (parties) in the 15-years-5-communities were defined as the identities of all individuals present in the party in 30-minutes time interval (cumulative parties)[83]. Given the fission-fusion dynamics of the bonobo social system, the sex-ratio of the daily parties, which potentially directly affects the dominance dynamics between the sexes, might be different from the ones of the communities. However, in the 15-years-5-communities dataset, the percentage of males (≥10 years) within the subgroups correlated strongly with the overall percentage of males (≥10 years) of the study communities (Spearman's rank correlation rho = 0.81, $P < 0.001$; SI Fig. 2). Parties in Bompusa West had a median of 4 males (Range= 0–9) and 5 females (Range 0–11), in Bompusa East of 3 males (Range= 0–7) and 4 females (Range= 0–11), in Ekalakala of 2 males (Range= 0–3) and 5 females (Range= 0–8), in Kokoalongo of 3 (Range= 0–10) males and 5 females (Range= 0–14) and in Fekako of 3 males (Range= 0–4) and 2 females (Range= 0–3).

### Data collection

At all sites, all occurrences of contact and non-contact aggressions were recorded during nest-to-nest follows of focal parties[60,84]. BF and GH collected behavioural data at Lomako and trained MS in data collection at LuiKotale. Subsequently, MS implemented at Kokolopori the data collection protocol from LuiKotale. We refer to conflicts as instances of dyadic agonistic interactions including an aggressor and a victim. Winning a conflict refers to an outcome in which an act of aggression elicits submissive behaviour in the victim. Coalitionary aggression refers to instances of agonistic interactions that include more than a single aggressor. Female coalitionary aggression in bonobos often incites participation and free-riding of additional individuals[46], sometimes even males (28% of the observed female coalitions in this study). We therefore scored each coalition involving more than one female in the role of primary aggressor as female coalition.

### Measurements of the degree of female power over males

We used two different measures to quantify the degree of female power over males, (1) the proportion of intersexual conflicts where males showed a submissive reaction to female aggression, and (2) the proportion of males in a community outranked by each female.

First, we used the *proportion of intersexual conflicts where males showed a submissive reaction to female aggression (referred to as: proportion conflicts won by females)* to quantify the degree of female power over males in a given community and year in the 30-years-6-communities

dataset. Due to several years with relatively few recorded agonistic interactions in certain communities, we miss information for many dyads (SI Table 1). However, this measurement of the degree of female power over males has been shown to correlate well with other indices based on dominance matrices[19]. We classified a conflict as won by the female if the male showed a submissive reaction including fleeing, moving aside or walking away.

Second, we used the *proportion of males in a community outranked on average by each female* based on David's scores (FDI-DS), to quantify the degree of female power over males in the 15-years-5-communities dataset, in addition to the proportion of conflicts won by females. Unlike the proportion of conflicts won by females, the FDI-DS considers the identity of the interacting individuals and is based on all decided agonistic interactions in the community (i.e., inter- and intrasexual conflicts).

The measurements of inter- and intrasexual power based on all interactions between all adults have been shown to correspond well to the ones based on either inter- and intrasexual interactions only[19,85]. Because bonobos lack a reliable, frequently used formal signal of subordinance status[86], we used only fleeing upon aggression as a marker of power to construct the interaction matrices[38].

### Measure of synchrony of sexual swellings

To quantify the synchrony of sexual swellings or the amount of time that multiple maximal tumescent females are present and the respective males' monopolization potential in a given year, we used the percentage of observation days on which two or more maximally tumescent females were observed during the party follow[87]. These data were only available for a subset of the communities ($N = 5$) and years ($N = 15$), during which a bonobo party was generally followed full days with a comparable data collection protocol (see SI Table 1).

### Measure of female coalitionary tendencies

We used two measures to quantify female coalition formation. Firstly, we used the frequency of all cases of female coalitionary aggression in a given community and year to quantify the propensities for these females to form coalitions. For that, we divided the observed number of female coalitionary aggression by the number of observation days. Secondly, we used the frequency of female coalitionary aggression against male targets in a given community and year to quantify the propensities for these females to form coalitions specifically against males. We therefore divided the observed number of female coalitions against males in a given community and year by the number of respective observation days. This measure captures the actual number of female coalitions in a given community and year, without controlling for the number of females in the community. Given female coalitions are hypothesized to suppress male aggression, the absolute number of occurrences of coalitionary aggression that an individual male receives from females is considered the relevant measure. These data are only available for a subset of the communities and years, during which bonobos were followed entire days, from nest-to-nest, with a comparable data collection protocol (see SI Table 1).

### Statistical analysis

To test the prediction of the **self-organization hypothesis**, we used the 30-years-6-communities and the 15-years-5-communities datasets investigating the link between the proportion of males within a community and the variation in the degree of female power over males.

To test the **reproductive control hypothesis**, and the **female coalition hypothesis**, we used the 15-years-5-communities dataset for which we have complete data on female swelling scores and all-occurrence observation of female coalitionary aggression (see SI Table 1). Because including the proportion of males into the same statistical models would have most likely led to overparameterization in the fixed effect part, we used this reduced dataset in a first step to test only the predictions of the female coalition hypothesis and the reproductive control hypothesis. In a second step, we verified the initial findings concerning the **self-organization hypothesis** in the smaller

15-years-5-cummunities dataset in a model including only the significant predictors of the initial model testing the reproductive control and the female coalition hypothesis.

MODEL 1. Testing the **self-organization hypothesis**: To test whether the proportion of males in a community is positively correlated with the degree of female power over males in bonobos (measured as the total percentage of intersexual conflicts where males showed a submissive reaction to female aggression), we fitted a Generalized Linear Mixed Model (GLMM) using binomial error structure and logit link function and the optimizer "bobyqa". As response variable we used a two-columns matrix with number conflicts in which the female was submissive to the male and the number of conflicts where the male was submissive to the female, respectively[88]. We included the predictor variable of proportion of males in the community as a fixed effect and included community and an observation level as a random intercepts effect. The inclusion of an observation level, which is basically a unique number for each observation year and community accounts for the potential non-independence of outcomes of conflicts observed in the same combination of community and year. This model was fitted using the 30-years-6-communities dataset. To evaluate potential temporal autocorrelation of data from the same community, we plotted the model residuals for each community for consecutive years. Positive autocorrelation would manifest in similar residuals in consecutive years. A visual inspection of the plot did not indicate autocorrelation (SI Fig. 3). In addition, for all communities with data for at least 4 consecutive years, we calculated an autocorrelation coefficient of the model residuals for each community, comparing one year to the consecutive one. The average autocorrelation per community was 0.004 (Spearman's correlation), revealing no indication of autocorrelation issues. To keep type I error rate at the nominal level of 0.05, we included a random slope of the proportion of males within the community[89,90]. We fitted the models in R (version 4.2.2; (R Core Team, 2022)) using maximum likelihood and the function glmer of the package lme4 (version 1.1-31[91]). We assessed the significance of the predictor variable using a likelihood ratio test (LRT[92]) comparing the full model and a reduced model comprising all the variables except the predictor variable of proportion of males[92]. The sample size consisted of 30 combinations of community and year; for 6 communities; females received submission from males in 1099 conflicts and females submitted to males in 687 conflicts.

To test for a potential influence of the proportion of subadult males (≥10 years and < =15 years) on the degree of female power over males in bonobos, we ran Model 1 with the predictor variable of proportion of subadult males as a fixed effect instead of proportion of males.

MODEL 2 (A/B/C) & 3 (A/B/C). Testing the **reproductive control hypothesis**, and the **female coalition hypothesis**: To test whether the synchrony of female sexual swellings (measured as the percentage of time at least two maximally tumescent females were present in the community; 2MTF), and the frequency of all female coalitions (FFC), influenced the degree of female power over males in bonobos, we fitted a series of GLMMs including these two predictor variables as fixed effects and social community and an observation level (a unique number for each observation year in each community; only in model 2) as a random effect. We fitted the model twice with two different ways to quantify the degree of female power over males as response variables: Model 2 (intersexual conflicts won by females) was fitted with the function glmer of lme4 using binomial error structure and logit link function and the optimizer "bobyqa", using two-columns matrix with number conflicts in which the female was submissive to the male and the number of conflicts where the male was submissive to the female, respectively, as response variable. Model 3 (Female power index based on David Score [FDI DS]) was fitted with the function glmmTMB of the package glmmTMB (version 1.1.5[93]) using a beta error structure and logit link function. As response we used the FDI DS, transformed such that values being exactly 0 or 1 are ruled out[94]. These models 2 and 3 were fitted with the 15-years-5-communities dataset. To avoid an overconfident model and keep type I error rate at the

nominal level of 0.05 we once included a random slope of our measurement of synchrony of female sexual swellings (2MTF; Models 2 A and 3 A) and once of the frequency of female coalition formation (FFC; Models 2B and 3B) into each model[89,90]. The reason for not including both random slopes in the same model was that for two of the five communities we had only two observations years in the data and for one community only three observation years. Hence, including both random slopes into the same model would have most likely led to a model being overparameterized in the random effects part. For the same reason we also did not include parameters for the correlation between the random intercept and slope. To avoid 'cryptic multiple testing'[95] and keep type I error rate at the nominal level of 0.05 we compared both of these full models with a respective null model lacking the fixed effects of FFC and 2MTF (but being otherwise identical), utilizing a likelihood-ratio test to obtain P-values for the individual fixed effects we dropped them from the fixed effects part, one at a time, and compared the resulting reduced models with the respective full model using an LRT (R function drop1). Since the predictor variable FFC was very skewed, also after log transformation, we used a variant of the Box-Cox transformation, allowing also the zeros in the variable to be transformed[96]. We implemented this transformation with the function powerTransform of the package car (version 3.0-12[97]). Prior to fitting the models, we z-transformed FFC and 2MTF to a mean of zero and a standard deviation of one to ease model convergence. For both models we estimated model stability by excluding individual communities and also individual combinations of community and year (only model 2) one at a time, fitting the respective full model to the obtained subset, and finally comparing the range of model estimates obtained for the subsets with those obtained for the full data set. This revealed both models to be of good stability. The models did not suffer from collinearity (maximum Variance Inflation Factor: 1.2[98]; function vif of the package car).

To evaluate potential temporal autocorrelation of data from the same community in subsequent years, we plotted the model residuals for each community in consecutive years. A visual inspection of the plot did not indicate autocorrelation (SI Fig. 3). For all communities with data in at least 4 consecutive years, we calculated an autocorrelation coefficient of the model residuals for each community, comparing one year to the consecutive one. The resulting average autocorrelations based on Spearman's correlation per community were 0 (Model 2 A), −0.25 (Model 2B), −0.25 (Model 3 A), 0.25 (Model 3B), revealing no indication of autocorrelation issues. We estimated confidence intervals of model coefficients and fitted values by means of a parametric bootstrap ($N = 1000$ bootstraps; function bootMer of the package lme4 or function simulate of the package glmmTMB).

Further, we re-ran all 4 models (Model 2 A/B and model 3 A/B) with the frequency of only the female coalition that were directed against males (FFCm), instead of the frequency of all female coalitions (FFC; Model $2A_{ff \to m}/2B_{ff \to m}/3A_{ff \to m}/3B_{ff \to m}$). Males might only consider female coalitions as threats when they are directed at males and we wanted to rule out that a slightly different quantification of the propensity of females to form coalitions affected the results of this study.

To test for the possibility that the proportion of males might influence one of the two measures of the degree of female power over males in the 15-years-5-communities dataset, we rerun the Models 2 and 3 with only the significant predictors and the proportion of males as additional predictor (Model 2 C/3 C; SI Table 2).

We have complied with all relevant ethical regulations for animal use

## Reporting summary
Further information on research design is available in the Nature Portfolio Reporting Summary linked to this article.

## Data availability
The data used for the formal analysis are provided in the supplemental information.

**Article**

## Code availability
The codes for the formal analysis are provided in the supplemental information.

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

## Acknowledgements
We thank the Institut Congolais pour la Conservations de la Nature and the Ministry of Scientific Research and Technology in the Democratic Republic of the Congo for their support and permission to work in the Kokolopori Bonobo Reserve, Democratic Republic of the Congo, as well as the Bonobo Conservation Initiative and Vie Sauvage. Thanks also to the Ministère de l'Education Nationale, Republique Democratique du Congo for granting permission to collect data at Lomako and to the Institut Congolais pour la Conservation de la Nature for granting research permissions to the LuiKotale Bonobo Project. Funding for Kokolopori was provided by Harvard University and the Max Planck Society. Funding for LuiKotale was provided by the Max Planck Society, the Centre for Research and Conservation of the Royal Zoological Society of Antwerp, Antwerpen, and The Leakey Foundation. We thank Charlotte Hemelrijk for her critical feedback throughout the manuscript's development process, as well as two anonymous reviewers from this, and two from an earlier version of the manuscript.

## Author contributions
Conceptualization: M.S.; Methodology: R.M., G.G., M.S.; Resources: M.S., G.H., B.F.; Data Curation L.C., G.G., M.K., G.H., B.F., M.S.; Investigation: M.S.; Writing—Original Draft: M.S.; Writing—Review and Editing: L.C., G.H., B.F.

## Funding

## Competing interests
The authors declare no competing interests.
