## [Transparent Peer Review file · Communications Biology]

DRIVERS OF FEMALE DOMINANCE IN BONOBOS

Corresponding Author: Dr Martin Surbeck

Version 0:

Reviewer comments:

Reviewer #1

(Remarks to the Author)

The paper addresses female dominance over males, a relatively rare occurrence in mammals. The authors test three different hypotheses proposed to explain female dominance over males, using bonobos as a case study. The paper sets up the problem well, presenting the three hypotheses, and summarising existing supporting evidence in mammals, for each. Each of the hypotheses generate contrasting predictions about under which conditions female dominance over males should arise which the authors then test using data from 6 communities of bonobos. The authors determine that of the three hypotheses, only one – that of female coalition formation, can explain the patterns of female dominance seen in their study.

I was excited to read this paper, it's an interesting topic that still leaves a lot to be understood. The paper was well structured, and I particularly like the way the three contrasting hypotheses and related predictions are set out and provide the structure throughout the MS.

Whilst the idea that female coalitions contribute to establishing female dominance over males is not a novel one, to my knowledge this is the first paper to empirically test between contrasting hypotheses in this way and presents an important step in determining the mechanisms for this characteristic, at least in this species. The findings are likely to be of interest to primatologists, biological anthropologists and those interested in evolution of social structures in mammals. It is not only the support for the female-coalition hypothesis that will be of interest, but also the lack of support found for the reduced contest hypothesis that will influence thinking and future avenues of study in this area.

The evidence to support the claims is sound, the dataset is good, especially for a large, long-lived mammal/great ape species, and includes multiple communities, populations and several years. The analytical approach is thorough. The manuscript is generally well-written and clear, some minor suggestions/revisions are outlined below.

Detailed comments with relevant line number below:

Line 40: delete 'can'.

Line 42: change 'to get' to 'of getting'.

Line 55: delete 'very'.

Line 58: add '(Macaca spp.)' for consistency.

Lines 85-86: italicise species names.

Line 90: Delete 'as well'.

Lines 168 – 175: Explain the particular relevance of the correlation (or not) of party composition to community composition, to this analysis/study. It is mentioned in the discussion (lines 516-523) but it would be helpful to make it explicit here.

Lines 199-203 (and again in lines 269-270): I have some trouble equating % observation days on which two or more maximally tumescent females are observed within a party, with the phrase 'synchrony of sexual swellings' which to me suggests wider synchrony across the community. For example, two females out of a community of 10 females at any one time wouldn't indicate a high overall degree of synchrony within the community, wouldn't something like proportion of females from the community that are maximally tumescent at any time be a better measure? I suggest removing the term 'quantify the synchrony of sexual swellings' and just keeping the actual measure you use, and/or including a statement as to why the measure used is the best indicator of synchrony, in the context of this analysis.

Lines 203-205: Including the sample size here in the text would be better than the reader having to go work it out from the supplementary info.

Line 226: typo – should be 'communities'.

Lines 269-270: see previous comment re measure of synchrony of female sexual swellings.

Results: This section does a good job of presenting the findings from a fairly complex set of analyses.

Lines 328 – 343: This is a nice initial summary of the patterns of dominance.

Line 370 (Table 1 legend) typo: 'overview of' not over.

Line 377: This sentence suggests two measures, (synchrony of swellings and presence of 2MF), rephrase to make clear that there is only one measure.

Line 409: Check the values for Model 2A Estimate and SE, think there are typos here.

Discussion:

This is a nice discussion of the findings in relation to the three hypotheses and relevant literature. It is well structured and easy to follow. Two additional points I'd like to see discussed/mentioned are:

i) The propensity for coalitions to be purely female-female versus mixed sex. The distinction is mentioned in the methods, and it's explained that both are included and why, but the question is left as to how many coalitions are only female-based, and whether looking only at those might make any difference to the findings.

ii) I think it is surprising that female tumescence/synchrony doesn't seem to influence the degree and patterns of female dominance and would like to see this discussed a little more. Is it possible that the measure used (2 or more in party) isn't sufficient to capture the degree of synchrony within the community and that it might be worth exploring this more in future? Perhaps around lines 529-549.

Lines 554 and 558: italicise species names.

Line 564 – The result that males were the targets of 85% of female coalitions would be good to see in the opening paragraphs of the results section where you summarise the outcomes (lines 328-335).

Line 596: specify the aspects of alliance formation you are referring to and 'modifies' should be 'modify'.

Reviewer #2

(Remarks to the Author)

This manuscript examines some of the factors that lead to females having power over males. The authors test three hypotheses using long-term data on bonobos from multiple field sites. They find that the best predictor of female intersexual power is female-female coalitions against males. I am very enthusiastic about this manuscript. It has the potential to be a very important paper and will be of interest to a wide audience. However, the authors need to address a few key issues if they want to have much of an impact.

1. Word choice. Science requires clear writing so that readers can fully understand what the authors are trying to communicate. In addition to some minor but important editing (see below), the authors need to avoid use interpretive terms when simpler and more straight forward terms are available. This will reduce reader confusion and allow the reader to more accurately evaluate what the authors are trying to convey. The authors are to be commended for defining some of these terms, but it is simpler if they just say what they mean.

a. "Dominance": The authors start the paper by contextualizing their research within the Lewis power framework but then quickly either abandon the terminology of that framework or misapply it. Because the authors do not define dominance but start by talking about the power framework, the reader assumes that the term is defined as in that framework (power based upon fighting ability), but then dominance is used as synonymous with power throughout. Importantly, that power framework, if the full set is included (which are mostly all cited in the paper: Lewis 2002, 2020, 2022 – the 2020 paper is missing but would be helpful: Female Power: A New Framework for Understanding "Female Dominance" in Lemurs), specifically talks about how problematic it is to conflate the terms dominance, power, rank, dominate, dominant, dominance rank, dominance status, etc. Below I have helped the authors apply the framework terminology correctly. The issue is of applying that framework's terminology is only partly an issue of the application, it is mostly that by using a term like dominance throughout (it is used 131 times, or more than 150 is you include dominant and dominate) without defining it and using it in different contexts, the authors' research becomes confusing and difficult to follow. If the authors decide to abandon the Lewis power framework, they need to at least be very careful and purposeful in their use of terms and to define exactly what they mean. The dominance literature is very confusing and conflicted in the use of this term. This paper will be most impactful if the authors state what they mean rather than the reader making assumptions.

b. "Won": The authors define winning as receiving a submissive reaction (line 186) but then talk about winning throughout. Receiving submission is just one of several definitions of winning that are out there in the dominance literature. It would be much clearer if they just said that they evaluated whether females have power over males by seeing if males are submissive to females in a conflict. If they want to take that interpretive step and call receiving submission as "winning", then they should do that in the Discussion. For the Methods and Results sections, at a minimum, the authors should state what they actually examined without interpretation, which is male submission to females.

c. "Rank": The authors conflate structural power (a hierarchy) with dyadic or triadic power when they state "proportion of males dominated" by females (e.g., line 188). The authors seem to be talking about the number of males that a female outranks in a hierarchy. The authors should be clear and about when they are talking about power structure and when they are talking about power relationships.

d. "Contest": The authors label one of their hypotheses the "reduced contest hypothesis" (e.g., line 77) but there is no need to bring in the concept of contest competition. Is the idea that the male-male contest competition is lower? Or is it that males are not in as much contest competition with the females? I think the idea is that the males are less aggressive than they might be because females have leverage (economic power) over the males due to control over reproduction. Contest competition is also jargon that is very field specific. By using terms that have less interpretation and jargon, the paper will be clearer to more readers. Instead of talking about contest throughout, the authors should state the simpler and clearer "reduced male aggression" or "reduced male aggression due to female control over reproduction" or if the authors use the power framework terms "reduced male aggression due to female leverage". If the authors want to connect it to their reference number 8 and the

discussion of contest vs scramble competition, I recommend doing that interpretive work in the Discussion. In the Discussion (lines 550-9), the authors talk about female reproductive autonomy under the subheading about synchrony of swellings. Also, maybe because I had to stop reading the manuscript part way through and pick it up again, but I had to go back and look at the Introduction to try to figure out which hypothesis this autonomy fit under because the term “reduced contest” doesn’t necessarily imply increased autonomy, and increased autonomy doesn’t necessarily mean reduced aggression (it could be just ineffective aggression). So, again, helping the reader keep track of what is actually being talked about (or known) and what is interpretation will help the reader follow the expectation, the findings, and the arguments.

2. Methods. One of the strengths of this manuscript is that it avails itself of 6 different datasets. By combing all of these data, the authors are able to test some factors that they couldn’t if they were using just 1 of the datasets. That being said, combining datasets can be problematic too. Were data collected in the same way? Did researchers define behaviors/events the same way? They obviously can’t do interobserver reliability tests, although it seems that there was some “cross-pollination”. However, the authors provide almost no information about the methods for how the data in the datasets were collected, just that they were compiled. Did they use focal animal sampling? Was it full-day follows? They list the number of days observed, was there a minimum number of hours to be considered a “day”? How many hours of observation per community and how were those hours distributed across individuals? How was a conflict defined? A coalition? Was it the same at all sites? They state that they included all males 10 yrs and older. What about females? Or do females only immigrate when they are adult? How did they define a subgroup? Was the distance requirement the same for all studies? It is ok if the datasets are not 100% compatible – we all recognize that there is variation out there in these methods. The combined dataset is incredible. But the reader needs to be able to evaluate it completely. We find out eventually in line 578 that 2 of the protocols are similar but that is too late and not with enough detail.

3. Comparative context. The authors compare their results with chimpanzees and the Wamba bonobo population but they do not discuss their results in light of the vast literature on “female dominant” species. These are the results for the drivers of female power in bonobos, how does that compare with what people have found in lemurs? In hyraxes? Their reference number 32 is all about how coalitions are what drive female power in spotted hyenas. They briefly mention spotted hyenas in a few places, but the Discussion section is mostly about bonobos or bonobos vs chimpanzees rather than a deep engagement with the literature of other species where people have tested overlapping ideas. Their reference number 24 finds sex ratio as predictive of female power and they test different social levels where sex ratio might be relevant. They also test reproductive control. These are just 2 studies that overlap extensively with the current paper. But there are more. This broader comparative context is particularly important given some of the arguments being made. For example, the authors seem to connect female power with reduced sexual coercion by males (though the authors don’t use that term) and yet forced copulation does occur in lemurs even when females are more powerful than males.

4. The big picture. The Discussion is mostly about bonobos. The authors test some big ideas with their amazing bonobo dataset. Yes, as a reader, I want the authors to tell me what it means for bonobos, but I also want to know what it means more broadly. What does this research, combined with all of the other amazing research out there (or is it too lacking? In which case, say what is lacking), what seems to be the drivers of intersexual power that is biased towards females? Does it seem to be different in each species? Is there a bigger pattern that might be emerging? The authors shouldn’t say more than they can, I’m not asking for speculation. Just a paragraph or so telling people who do not study bonobos what this paper means for understanding animal behavior more broadly. The authors have such an interesting paper, it would be great if they would make it more relevant to the broader community.

In addition, some minor issues need to be addressed, including the numerous typos and the frequent long, convoluted sentences that can be hard to follow. Additionally, the overuse of weak sentence structure is problematic. The authors rely heavily on the sentence structure that follows the pattern “There is ...” or “There are...” or “It is...” without “it” referring to any noun. Occasional use of this structure is ok, but this structure can be less clear for the reader. Make the actor the subject of the sentence: Instead of “It seems plausible that the general lower variation within communities arises from...”(line581-2), say “The general lower variation within communities may arise from”. Or instead of “There seems to be between-site variation in the triggers of female coalition formation...”(line585), say “The triggers of female coalition formation seem to vary between sites...”. Finally, I list minor issues below, as well as how to change the terminology to fit the power framework.

Line

27-30 sentence structure is awkward

37,51 While Ralls 1977 is commonly cited for saying that males are dominant to females in most mammals, she is very clear to state that many species hadn’t been studied at that time. The authors also cite her for saying that males are larger than females in most mammals, but a new paper (Ralls is 50 years old) contradicts the dimorphism statement and leads one to wonder if Ralls’ often forgotten qualification about intersexual dominance might also fall with more data. Tombak et al (2024) state “Our analyses of wild, non-provisioned populations representing >400 species indicate that although males tend to be larger than females when dimorphism occurs, males are not larger in most mammal species, suggesting a need to revisit other assumptions in sexual selection research.”

Tombak, K.J., Hex, S.B.S.W. & Rubenstein, D.I. New estimates indicate that males are not larger than females in most mammal species. *Nat Commun* 15, 1872 (2024). <https://doi.org/10.1038/s41467-024-45739-5>

rank, status, dominance, power? Which one? If the authors are including leverage, which they include in the next sentence (line 42), then it should be rank, status, or power without reference to dominance. See reference 7 or Drews 1993 (cited therein) for a distinction between the terms “status” and “rank”. Status is dyadic and rank is group level, so it seems status is more appropriate than rank in this sentence.
use power instead of dominance here
I think the authors mean intersexual power here instead of only power based on fighting ability (dominance)
use power instead of dominance here
use power instead of dominance here
No citations for this assumption. There are other causes of sexual dimorphism. Adam Gordon has multiple papers talking about sex-specific body size responses to ecology.
use power instead of dominance here
use power instead of dominance here
"This" is a descriptor and requires a noun.
change to "are more powerful than" or "have more power than"
use power instead of dominance here
use power instead of dominance here
"This" is a descriptor and requires a noun
change "dominance rank" to "rank" or "position in the hierarchy" or similar to avoid limiting rank to only the position in the hierarchy based on fighting ability
use power instead of dominance here. Reference number 24 showed that leverage due to market effects can fluctuate with sex ratio. A comparison of these ideas about sex ratio should occur somewhere in the manuscript.
Change the name of this hypothesis to "leverage" or "reproductive control" to remove some of the jargon and additional layer of interpretation (see above).
Add to just after "is reduced" the following: "by being aggressive towards females"; "This" is a descriptor and requires a noun
Why are the ranks based on dominance when on line 81 the authors state that it is leverage? Rephrase: "and female position in the power hierarchy rise relative to males"
use power instead of dominance here
Insert "hypothesis" between "this" and "is" and delete on line 89
use power instead of dominance here
"This" is a descriptor and requires a noun
sister species is understood given that they are in the same genus
103-4 Rewrite: Although the high degree of female power over males may be a derived trait (Lewis et al 2023), in bonobos, the actual underlying mechanisms are still misunderstood.
The statement needs a citation, I recommend: Lewis RJ, Kirk EC, Gosselin-Ildari AD. Evolutionary Patterns of Intersexual Power. *Animals*. 2023; 13(23):3695. <https://doi.org/10.3390/ani13233695>
use power instead of dominance here
use ranking higher instead of dominating here
use power instead of dominance here
use power instead of dominance here
use power or leverage instead of dominance here
use power or leverage instead of dominance here
use power instead of dominance here
for clarity, replace "as" with "because"
130-7 The authors use the term dominance correctly within the power framework here. They may want to use the general term "power" for consistency, but it fits the framework either way.
138,144 Here and throughout, inconsistency in whether numbers are spelled out or not
Methods & Results: averages should have standard deviations or standard errors and ranges should be associated with medians. Decimal places are inconsistent and not biologically meaningful. Line 161 community size was 16.67 individuals – what is 1/100th of an individual?
170-2 include the variation in these numbers
use power instead of dominance here
use power instead of dominance here
use outranked instead of dominated here
use power instead of dominance here
use power instead of dominance here
186-7 How were these behavioral data collected across sites? Focal? All occurrences for all individuals? How operationalize "conflict" "coalition" etc.
use outranked instead of dominated here
use power instead of dominance here
use power instead of dominance here
Use "because" instead of "as"
use power instead of dominance here
delete "dominance"
change "are" to "were"
aggression is usually singular and plural
does this mean that sometimes the female coalitions include males as well? Mixed-sex?
use power instead of dominance here
Use "because" instead of "as"
use power instead of dominance here
Change to "where males showed a submissive reaction to" or something similar
Change to "conflicts in which the female was submissive to the male and the number of conflicts where the male was submissive to the female"
262-3 Change to "males submitted to females in 1099 conflicts and females submitted to males in 687 conflicts"
265 use power instead of dominance here

MF as a code for maximally tumescent females is confusing for readers who typically think of MF as male/female or multifemale, maybe MTF?
use power instead of dominance here
what is observation level?
use power instead of dominance here
rephrase "won" to refer to submission
only 2 observations...??? Of what? Also, what can really be said about only a handful of observation over multiple years?

316 re-run? Re-ran?

318-21 This sentence is confusing.

323 use power instead of dominance here; rerun? Reran? Re-ran?

328 Bonobo females often had power over males

329-30 use power instead of dominance here; change to "...intersexual conflicts in which males submitted...dataset, male bonobos submitted to females on average in 61%..."

Change to "percentage of intersexual conflicts in which males submitted to females with ..."; this is HUGE variation! Was the submission to just 1 female and not a coalition of females? This result needs to be better addressed explicitly.
use power instead of dominance here
delete "dominance"
Change to "...measures of female power over males, the proportion of conflicts in which males submitted to females..."
use power instead of dominance here
Change to "conflicts where females received submission from males within a community..."
use power instead of dominance here
Rephrase "won" to refer to submission
use power instead of dominance here

Figure 1 y-axis: change to "proportion of intersexual conflicts in which males submit"

delete "degree of female dominance over males measured as the"
Change "won" to "in which males submit to females"
use power instead of dominance here
Rephrase "won"
use power instead of dominance here
Rephrase "won"
use power instead of dominance here
use outranked instead of dominated here
weak evidence? The result is not significant!
little evidence? The result is not significant!
use power instead of dominance here
Rephrase "won"
use power instead of dominance here; there are multiple decimals in these values.
use outranked instead of dominated here
use power instead of dominance here
moderate evidence based on $p=0.036$ and weak evidence based on $p=0.083$? The phrasing throughout the Results with weak or little or moderate evidence based on the p-value is concerning. Maybe plot effect size and talk about strong, moderate, or weak effects? If the authors want to talk about the direction of an effect was consistent with expectations even if it wasn't significant, I'm more ok with that than the current little/weak/moderate/strong language.
why spell out and not use the FFC abbreviation?
strong evidence based on p-value? See above.
use power instead of dominance here
calling a nonsignificant result as weak evidence is problematic
use power instead of dominance here

Figure 2 y-axis: y-axis: change to "proportion of intersexual conflicts in which males submit" and "David's Scores"

use power instead of dominance here
adjust text to fit the new y-axis title

Table 2 adjust text to fit new language

use power instead of dominance here

456-7 adjust text to fit new language

use power instead of dominance here
use power instead of dominance here
Change to "ranking higher than males"
Use "because" instead of "as"
use power instead of dominance here
"behavioral tool" is an odd word choice. The power framework would suggest that it is not a behavioral tool but just a different base of power. I don't think the authors need to apply the framework terminology here unless they want to, but I would phrase differently. Strategy?
use power instead of dominance here
which indices? There was the FDI-DS, what were the other indices? Or is this across communities? Make sure wording is consistent here "dominance indices"; How did the Wamba study define "won"?

481-2 awkward wording; change to "a male had the highest rank..."; was the mother present?

How often were males with mothers in this study? Is being with the mother a "behavioral tool" for a male?
use greater power instead of higher dominance here
use power instead of dominance here
relevance or consequence?
Change to "...a given female's individual rank..."
ownership?; change to "intersexual rank"; the sentence is confusing
delete dominance
already in the their process of reaching adulthood? Odd wording
so why is this a hypothesis? What about other "female dominant" species and coercive mating?
delete dominance
leadership is a completely different phenomenon and seems to come out of nowhere. Females in some "male dominant" species lead group movement.
use power instead of dominance here
typo
512-3 use power instead of dominance here; Do we know if there are winner/loser effects in these species? Do we know if male-male conflicts increase in frequency as male numbers increase? Key assumptions are not addressed.
The key assumptions should be addressed prior to this point and it should be discussed in the Intro about where bonobos fit and where they don't. Bonobos do live in permanent groups, just not cohesive groups. Is the assumption of DomWorld permanent groups or cohesive groups?
What is the relevant level for assessing sex ratio? How relate to the findings about sex ratios in reference number 24?
buildup? What does that mean?; use power instead of dominance here; and why is this statement true?
use power instead of dominance here
more than ?; more active versus a passive role?
use power instead of dominance here
use power instead of dominance here; needs a citation; this phrasing places the majority of the agency with males and females appear to be passive recipients without the ability to have any influence on outcomes. Do the authors mean to imply that females do not have much agency?
change to "a species for which females sexual swellings..."
why do the authors all of a sudden decide to use the term "power" here?
typo
see previous comment about status and whether it needs to have the descriptor "dominance" in front of it.
but lemurs still exhibit forced copulation despite "female power". It is not a zero-sum game with either males or females having complete autonomy. I think this discussion needs a little more nuance and discussion about the context of power.
Use "because" instead of "as"
also?; use "because" instead of "as"
"This makes it" What is "this" and what is "it"?
Use "because" instead of "as"
contraction
delete dominance
use power instead of dominance here
delete dominance
use power instead of dominance here
use power instead of dominance here
use power instead of dominance here
This needs a noun after it
use power instead of dominance here
use power instead of dominance here

Supplemental material

Table S1 is confusing:

Headings:

-what is agi?

-for observation days, what is 64 days for 20 individuals? How many hours of observations with each individual or each possible dyad? And how are there dyadic data without any days of observation?

-Female coalitions: any information on the response to the coalition?

-Do the authors mean % or proportion? All of the values are between 0 and 1.

842-4 use power instead of dominance here

845 use outranked instead of dominated here

850 adjust wording "wins"

852 adjust wording "won" and "dominated"

Table S2 & S3 adjust wording

Version 1:

Reviewer comments:

Reviewer #2

(Remarks to the Author)

The authors have done a pretty good job revising the manuscript. I have just a few comments, which are mostly copyedits rather than content. The manuscript needs some cleaning up: There are some missing periods at the end of sentences, extra

periods at the end of sentences, missing hyphens in hyphenated words, missing spaces at the ends of sentences, etc.

1. The authors said they addressed the sentences with weak structure but they still have numerous sentences, perhaps that were added in the revision and not caught?

Line 26, 63, 480 "This" is a descriptor and must be followed by a noun.

Lines 124, 297, 380, 381, 391, 413, 451, 452 "It" is a pronoun and must refer to a noun. Rephrase for improved clarity.

Lines 186, 206, 212, 237, 247 "there was..." needs to be revised for improved clarity.

Lines 287, 288, 296, 322 "there are..." needs to be revised for improved clarity.

Line 300, 304, 308, 354, 397 "there ..."

2. Verb tense.

Line 81 "rise" should not be in the present tense

Line 91 change "is" to "was"

Line 111 change "predict" to "predicted"

Line 127 change "expect" to "expected"

Line 135 change "predict" to "predicted"

Line 184 change "is" to "was"

3. Line 86, is it true that female coalition hypothesis is the least discussed? Are there any citations or something that can back up this assertion?

4. Line 113, "been [proposed] based on..." is this what the authors are saying? It seems the word proposed was missing.

5. Line 174, 1 sentence does not equal a paragraph.

6. Line 332-5, this sentence is awkward.

7. Line 390, typo: replace "where" with "were"

8. Line 409, no contractions in formal writing

9. Line 494, do you mean "elicit" (to get) or "solicit" (to request)?

10. Line 913, the figure caption should mention the A and B panels in the figure.

RESPONSE TO REFEREES

We want to thank both reviewers for the time and effort they invested into our manuscript

Find below the detailed replies to each inquiry in bold italics

Reviewers' comments:

Reviewer #1 (Remarks to the Author):

The paper addresses female dominance over males, a relatively rare occurrence in mammals. The authors test three different hypotheses proposed to explain female dominance over males, using bonobos as a case study. The paper sets up the problem well, presenting the three hypotheses, and summarising existing supporting evidence in mammals, for each. Each of the hypotheses generate contrasting predictions about under which conditions female dominance over males should arise which the authors then test using data from 6 communities of bonobos. The authors determine that of the three hypotheses, only one – that of female coalition formation, can explain the patterns of female dominance seen in their study.

I was excited to read this paper, it's an interesting topic that still leaves a lot to be understood. The paper was well structured, and I particularly like the way the three contrasting hypotheses and related predictions are set out and provide the structure throughout the MS.

Whilst the idea that female coalitions contribute to establishing female dominance over males is not a novel one, to my knowledge this is the first paper to empirically test between contrasting hypotheses in this way and presents an important step in determining the mechanisms for this characteristic, at least in this species. The findings are likely to be of interest to primatologists, biological anthropologists and those interested in evolution of social structures in mammals. It is not only the support for the female-coalition hypothesis that will be of interest, but also the lack of support found for the reduced contest hypothesis that will influence thinking and future avenues of study in this area.

The evidence to support the claims is sound, the dataset is good, especially for a large, long-lived mammal/great ape species, and includes multiple communities, populations and several years. The analytical approach is thorough. The manuscript is generally well-written and clear, some minor suggestions/revisions are outlined below.

We thank the reviewer for an overall very positive assessment of our manuscript and for the time and work invested to provide feedback. We outline below in detail how we addressed the different points raised by the reviewer 1.

Detailed comments with relevant line number below:

Line 40: delete 'can'.

Done

Line 42: change 'to get' to 'of getting'.

Done

Line 55: delete 'very'.

Done

Line 58: add '(Macaca spp.)' for consistency.

Done, also added (*Lemuroidea spp*) in sentence before for consistency (L52/55)

Lines 85-86: italicise species names.

Done

Line 90: Delete 'as well'.

Done

Lines 168 – 175: Explain the particular relevance of the correlation (or not) of party composition to community composition, to this analysis/study. It is mentioned in the discussion (lines 516-523) but it would be helpful to make it explicit here.

Done

We adjusted the phrasing of the paragraph in the methods and it reads now (L497ff):

“ Given the fission-fusion dynamics of the bonobo social system, the sex-ratio of the daily subgroups (parties), which potentially directly affects the dominance dynamics between the sexes, might be different from the ones of the communities. However, this was not the case in the 15-years-5-communities dataset, as the percentage of males (≥ 10 years) within the subgroups (parties) correlated strongly with the overall percentage of males (≥ 10 years) of the study communities (Spearman's rank correlation $\rho = 0.81$, $P < 0.001$; SI Figure 1). Parties in Bompusa West consisted on average of 4.1 males and 5.1 females, in Bompusa East of 3.1 males and 4.6 females, in Ekalakala of 2.3 males and 5.4 females, in Kokoalongo of 3.4 males and 5.5 females and in Fekako of 2.7 males and 2.4 females.”

Lines 199-203 (and again in lines 269-270): I have some trouble equating % observation days on which two or more maximally tumescent females are observed within a party, with the phrase 'synchrony of sexual swellings' which to me suggests wider synchrony across the community. For example, two females out of a community of 10 females at any one time wouldn't indicate a high overall degree of synchrony within the community, wouldn't something like proportion of females from the community that are maximally tumescent at any time be a better measure? I suggest removing the term 'quantify the synchrony of sexual swellings' and just keeping the actual measure you use, and/or including a statement as to why the measure used is the best indicator of synchrony, in the context of this analysis.

We are grateful to the reviewer to point this out and follow their suggestion and removed the term.

Lines 203-205: Including the sample size here in the text would be better than the reader having to go work it out from the supplementary info.

Following the suggestion of the reviewer, we included the sample sizes of number of communities (N=5) and years (N=15) (L548)

Line 226: typo – should be 'communities'.

Thanks for spotting this, corrected

Lines 269-270: see previous comment re measure of synchrony of female sexual swellings.

We hope this should be clear now with the insertion of sample size above

Results:

This section does a good job of presenting the findings from a fairly complex set of analyses.

Lines 328 – 343: This is a nice initial summary of the patterns of dominance.

We thank the reviewer for their appraisal

Line 370 (Table 1 legend) typo: 'overview of' not over.

Corrected

Line 377: This sentence suggests two measures, (synchrony of swellings and presence of 2MF), rephrase to make clear that there is only one measure.

Thank you for pointing this out, we modified the sentence to "synchrony of swellings QUANTIFIED BY the presence of 2MF."(L 210)

Line 409: Check the values for Model 2A Estimate and SE, think there are typos here.

Yes, thank you for spotting, we corrected the values in the revised version of the manuscript

Discussion:

This is a nice discussion of the findings in relation to the three hypotheses and relevant literature. It is well structured and easy to follow. Two additional points I'd like to see discussed/mentioned are:

i) The propensity for coalitions to be purely female-female versus mixed sex. The distinction is mentioned in the methods, and it's explained that both are included and why, but the question is left as to how many coalitions are only female-based, and whether looking only at those might make any difference to the findings.

To respond to this point of the reviewer we now specify the percentage of female coalition in our study which are accompanied by males in the method section: 28%, leaving 72% of female coalitions as only females (L 517). We further discuss whether the male participation in female coalitions potentially affects our findings: "In all populations we sometimes observed males participating in female coalitions. The participating males never were in leading positions but rather trailed behind several female aggressors (MS personal observation). It seems highly unlikely that their participation impacted observed dynamics and consequently affected our findings."(L408)

ii) I think it is surprising that female tumescence/synchrony doesn't seem to influence the degree and patterns of female dominance and would like to see this discussed a little more. Is it possible that the measure used (2 or more in party) isn't sufficient to capture the degree of synchrony within the community and that it might be worth exploring this more in future? Perhaps around lines 529-549.

We agree with the reviewer to be cautious about this negative finding. Particularly, as we have shown previously that patterns of female tumescence affect the dominance dynamic within male female dyads. We follow the suggestions of the reviewer and added in the discussion a part about how to might better quantify population differences in male monopolization potential (L 373): "We used the previously established measurement of percentage of days with more than two maximally tumescent females in a community to quantify the female monopolization potential by males and the potential relevance of male aggression. While this seems a good initial step, future

studies trying to quantify community differences in the relevance of male aggression should include additional measures, such as reliability of sexual swellings in each population or the responsiveness of male androgen levels to the presence of maximally tumescent females.”

Lines 554 and 558: italicise species names.

Corrected

Line 564 – The result that males were the targets of 85% of female coalitions would be good to see in the opening paragraphs of the results section where you summarise the outcomes (lines 328-335).

Added: “In 85% of the cases, female coalitions mainly targeted males” to the mentioned opening paragraph of the result sections (L 236), as well as to the abstract

Line 596: specify the aspects of alliance formation you are referring to and ‘modifies’ should be ‘modify’.

Corrected and added composition and frequency as aspects that we incorporated in our study (L441)

Reviewer #2 (Remarks to the Author):

This manuscript examines some of the factors that lead to females having power over males. The authors test three hypotheses using long-term data on bonobos from multiple field sites. They find that the best predictor of female intersexual power is female-female coalitions against males. I am very enthusiastic about this manuscript. It has the potential to be a very important paper and will be of interest to a wide audience. However, the authors need to address a few key issues if they want to have much of an impact.

We are grateful to reviewer 2 for the time and effort to review our manuscript, for providing all the detailed comments, as well as for an overall positive evaluation.

1. Word choice. Science requires clear writing so that readers can fully understand what the authors are trying to communicate. In addition to some minor but important editing (see below), the authors need to avoid use interpretive terms when simpler and more straight forward terms are available. This will reduce reader confusion and allow the reader to more accurately evaluate what the authors are trying to convey. The authors are to be commended for defining some of these terms, but it is simpler if they just say what they mean.

a. “Dominance”: The authors start the paper by contextualizing their research within the Lewis power framework but then quickly either abandon the terminology of that framework or misapply it. Because the authors do not define dominance but start by talking about the power framework, the reader assumes that the term is defined as in that framework (power based upon fighting ability), but then dominance is used as synonymous with power throughout. Importantly, that power framework, if the full set is included (which are mostly all cited in the paper: Lewis 2002, 2020, 2022 – the 2020 paper is missing but would be helpful: Female Power: A New Framework for Understanding “Female Dominance” in Lemurs), specifically talks about how problematic it is to conflate the terms dominance, power, rank, dominate, dominant, dominance rank, dominance status, etc. Below I have helped the authors apply the framework terminology correctly. The issue is of applying that framework’s terminology is only partly an issue of the application, it is mostly that by using a term like dominance throughout (it is used 131 times, or more than 150 if you include

dominant and dominate) without defining it and using it in different contexts, the authors' research becomes confusing and difficult to follow. If the authors decide to abandon the Lewis power framework, they need to at least be very careful and purposeful in their use of terms and to define exactly what they mean. The dominance literature is very confusing and conflicted in the use of this term. This paper will be most impactful if the authors state what they mean rather than the reader making assumptions.

We are extremely grateful to the reviewer for their help in applying the framework of power adequately.

In the revised version of the manuscript, we are more careful in the application of the framework of power, which we think is useful in this context and follow the suggestion of the reviewer. We include the suggested citations.

b. "Won": The authors define winning as receiving a submissive reaction (line 186) but then talk about winning throughout. Receiving submission is just one of several definitions of winning that are out there in the dominance literature. It would be much clearer if they just said that they evaluated whether females have power over males by seeing if males are submissive to females in a conflict. If they want to take that interpretive step and call receiving submission as "winning", then they should do that in the Discussion. For the Methods and Results sections, at a minimum, the authors should state what they actually examined without interpretation, which is male submission to females.

We agree with the reviewer and revised the manuscript in a way that is more explicit about the actual outcome of conflicts. We use the term "intersexual conflicts in which males submitted to females" throughout the revised version of the manuscript

c. "Rank": The authors conflate structural power (a hierarchy) with dyadic or triadic power when they state "proportion of males dominated" by females (e.g., line 188). The authors seem to be talking about the number of males that a female outranks in a hierarchy. The authors should be clear and about when they are talking about power structure and when they are talking about power relationships.

We adapted the manuscript according to the reviewer's suggestions and changed the term "proportion of males dominated by females" to "average proportion of males in a community outranked by each female" throughout the manuscript.

d. "Contest": The authors label one of their hypotheses the "reduced contest hypothesis" (e.g., line 77) but there is no need to bring in the concept of contest competition. Is the idea that the male-male contest competition is lower? Or is it that males are not in as much contest competition with the females? I think the idea is that the males are less aggressive than they might be because females have leverage (economic power) over the males due to control over reproduction.

(The reviewer is correct in their assumption)

Contest competition is also jargon that is very field specific. By using terms that have less interpretation and jargon, the paper will be clearer to more readers. Instead of talking about contest throughout, the authors should state the simpler and clearer "reduced male aggression" or "reduced male aggression due to female control over reproduction" or if the authors use the power framework terms "reduced male aggression due to female leverage". If the authors want to connect it to their reference number 8 and the discussion of contest vs scramble competition, I recommend doing that interpretive work in the Discussion. In the Discussion (lines 550-9), the authors talk about female reproductive autonomy under the subheading about synchrony of swellings. Also, maybe because I had to stop reading the manuscript part way through and pick it up again, but I had to go back and

look at the Introduction to try to figure out which hypothesis this autonomy fit under because the term “reduced contest” doesn’t necessarily imply increased autonomy, and increased autonomy doesn’t necessarily mean reduced aggression (it could be just ineffective aggression). So, again, helping the reader keep track of what is actually being talked about (or known) and what is interpretation will help the reader follow the expectation, the findings, and the arguments.

According to the reviewer’s suggestion we changed the name to “reproductive control hypothesis” throughout the manuscript and we no longer refer to contest vs scramble competition. We further modified the text about female reproductive autonomy under the subheading about synchrony of swellings and included insights from the Sifaka (L85, L377).

2. Methods. One of the strengths of this manuscript is that it avails itself of 6 different datasets. By combing all of these data, the authors are able to test some factors that they couldn’t if they were using just 1 of the datasets. That being said, combining datasets can be problematic too. Were data collected in the same way? Did researchers define behaviors/events the same way? They obviously can’t do interobserver reliability tests, although it seems that there was some “cross-pollination”. However, the authors provide almost no information about the methods for how the data in the datasets were collected, just that they were compiled. Did they use focal animal sampling? Was it full-day follows? They list the number of days observed, was there a minimum number of hours to be considered a “day”? How many hours of observation per community and how were those hours distributed across individuals? How was a conflict defined? A coalition? Was it the same at all sites? They state that they included all males 10 yrs and older. What about females? Or do females only immigrate when they are adult? How did they define a subgroup? Was the distance requirement the same for all studies? It is ok if the datasets are not 100% compatible – we all recognize that there is variation out there in these methods. The combined dataset is incredible. But the reader needs to be able to evaluate it completely. We find out eventually in line 578 that 2 of the protocols are similar but that is too late and not with enough detail.

We extended the methods in the revised version of the manuscript accordingly. The revised version now includes 1) Precision of the sampling methods and how it got implemented across the sites (L509 ff); 2) definition of conflicts/winning/ coalitions L312ff)3) ages of included female and its rational (LL487); 4) definition of subgroups (L496). We do not see the relevance of having individual observation hours reported, as they do not seem to directly affect our measures of female dominance and therefore refrain from extracting and presenting these values.

3. Comparative context. The authors compare their results with chimpanzees and the Wamba bonobo population but they do not discuss their results in light of the vast literature on “female dominant” species. These are the results for the drivers of female power in bonobos, how does that compare with what people have found in lemurs? In hyraxes? Their reference number 32 is all about how coalitions are what drive female power in spotted hyenas. They briefly mention spotted hyenas in a few places, but the Discussion section is mostly about bonobos or bonobos vs chimpanzees rather than a deep engagement with the literature of other species where people have tested overlapping ideas. Their reference number 24 finds sex ratio as predictive of female power and they test different social levels where sex ratio might be relevant. They also test reproductive control. These are just 2 studies that overlap extensively with the current paper. But there are more. This broader comparative context is particularly important given some of the arguments being made. For example, the authors seem to connect female power with reduced sexual coercion by males (though the authors don’t use that term) and yet forced copulation does occur in lemurs even when females are more powerful than males.

In line with the suggestions of the reviewer we revised parts of the discussion to embed our findings in insights from other species including lemurs, vervet monkeys, capuchins and macaques. For example, we added the findings from Rock Hyraxes in our discussion of the self-organisation hypothesis (L338ff), we added findings in Verreaux's sifaka to our discussion of potential sources of female leverage and how changes in the availability of potentially receptive females affect the dynamic between the sexes (337ff), and we further elaborated on the findings in spotted hyenas in the section about female coalitions (L395ff). Finally, we now are careful to note that reproductive autonomy and female power are not always the same and that we observe cases of forced copulations and infanticide even in lemur species with high female power(L400).

4. The big picture. The Discussion is mostly about bonobos. The authors test some big ideas with their amazing bonobo dataset. Yes, as a reader, I want the authors to tell me what it means for bonobos, but I also want to know what it means more broadly. What does this research, combined with all of the other amazing research out there (or is it too lacking? In which case, say what is lacking), what seems to be the drivers of intersexual power that is biased towards females? Does it seem to be different in each species? Is there a bigger pattern that might be emerging? The authors shouldn't say more than they can, I'm not asking for speculation. Just a paragraph or so telling people who do not study bonobos what this paper means for understanding animal behavior more broadly. The authors have such an interesting paper, it would be great if they would make it more relevant to the broader community.

We extended the discussion and the conclusions to incorporate the findings of other species and try to give a bigger picture (L460ff).

In addition, some minor issues need to be addressed, including the numerous typos and the frequent long, convoluted sentences that can be hard to follow. Additionally, the overuse of weak sentence structure is problematic. The authors rely heavily on the sentence structure that follows the pattern "There is ..." or "There are..." or "It is..." without "it" referring to any noun. Occasional use of this structure is ok, but this structure can be less clear for the reader. Make the actor the subject of the sentence: Instead of "It seems plausible that the general lower variation within communities arises from..."(line581-2), say "The general lower variation within communities may arise from". Or instead of "There seems to be between-site variation in the triggers of female coalition formation..."(line585), say "The triggers of female coalition formation seem to vary between sites...". Finally, I list minor issues below, as well as how to change the terminology to fit the power framework.

We are grateful for the reviewer for undertaking all this work and we follow tightly the suggestions concerning the application of the power framework in the revised version of the manuscript. Furthermore, we changed a large portion of the "There is ..." sentences.

Line

27-30 sentence structure is awkward

We modified the sentence as we also shortened the abstract. It reads now (L26):" Our results only support predictions of the female coalition hypothesis. We find that females occupy higher dominance ranks compared to males when they form more frequent coalitions indicating that female coalition formation is a behavioral tool for females to gain dominance over males."

37,51 While Ralls 1977 is commonly cited for saying that males are dominant to females in most mammals, she is very clear to state that many species hadn't been studied at that time. The authors also cite her for saying that males are larger than females in most mammals, but a new paper (Ralls is

50 years old) contradicts the dimorphism statement and leads one to wonder if Ralls' often forgotten qualification about intersexual dominance might also fall with more data. Tombak et al (2024) state "Our analyses of wild, non-provisioned populations representing >400 species indicate that although males tend to be larger than females when dimorphism occurs, males are not larger in most mammal species, suggesting a need to revisit other assumptions in sexual selection research."

Tombak, K.J., Hex, S.B.S.W. & Rubenstein, D.I. New estimates indicate that males are not larger than females in most mammal species. Nat Commun 15, 1872 (2024). <https://doi.org/10.1038/s41467-024-45739-5>

In line with the reviewers' suggestion, we are now careful to cite the Ralls paper only for the traditional statement about male dominance, which we update further in the introduction. In the revised version we use in the modified statement about body size the Tombak et al Citation (Figure 2 in this publication still shown that the majority of primates show a male bias in body mass dimorphism, so the statement about primates hold true).

rank, status, dominance, power? Which one? If the authors are including leverage, which they include in the next sentence (line 42), then it should be rank, status, or power without reference to dominance. See reference 7 or Drews 1993 (cited therein) for a distinction between the terms "status" and "rank". Status is dyadic and rank is group level, so it seems status is more appropriate than rank in this sentence.

We are grateful to the reviewer to invest the time to straighten the nomenclature in line the concept of power. We follow the suggestions throughout the manuscript.

use power instead of dominance here

Done

I think the authors mean intersexual power here instead of only power based on fighting ability (dominance)

Done

use power instead of dominance here

Done

use power instead of dominance here

Done

No citations for this assumption. There are other causes of sexual dimorphism. Adam Gordon has multiple papers talking about sex-specific body size responses to ecology.

Citation added

use power instead of dominance here

Done

use power instead of dominance here

Done

"This" is a descriptor and requires a noun.

Added "variation"

change to "are more powerful than" or "have more power than"

Done

use power instead of dominance here

Done

use power instead of dominance here

Done

"This" is a descriptor and requires a noun

added “variation”

change “dominance rank” to “rank” or “position in the hierarchy” or similar to avoid limiting rank to only the position in the hierarchy based on fighting ability

Done

use power instead of dominance here.

Done

Reference number 24 showed that leverage due to market effects can fluctuate with sex ratio. A comparison of these ideas about sex ratio should occur somewhere in the manuscript.

We thank the reviewer for pointing this out and added in the introduction of the revised version of the manuscript the following statement (L73ff): “. An alternative explanation for the link between sex-ratio and female power within a group might be that a decreasing female biased sex-ratio changes the supply and demand for mating opportunities and reduces the leverage of females”²⁵

Change the name of this hypothesis to “leverage” or “reproductive control” to remove some of the jargon and additional layer of interpretation (see above).

We are grateful for this suggestion and changed the name of the hypothesis to “reproductive control hypothesis” (L76) and changed the name throughout the revised version of the manuscript

Add to just after “is reduced” the following: “by being aggressive towards females”; “This” is a descriptor and requires a noun

Added the relevant term(L79): “reduction in the ability to monopolize potentially fertile females”

Why are the ranks based on dominance when on line 81 the authors state that it is leverage? Rephrase: “and female position in the power hierarchy rise relative to males”

Done

use power instead of dominance here

Done

Insert “hypothesis” between “this” and “is” and delete on line 89

Done

use power instead of dominance here

Done

“This” is a descriptor and requires a noun

added “dynamic between the sexes”

sister species is understood given that they are in the same genus

we would like to leave this emphasis in the manuscript, even though it is a bit redundant

103-4 Rewrite: Although the high degree of female power over males may be a derived trait (Lewis et al 2023), in bonobos, the actual underlying mechanisms are still misunderstood.

The statement needs a citation, I recommend: Lewis RJ, Kirk EC, Gosselin-Ildari AD. *Evolutionary Patterns of Intersexual Power. Animals. 2023; 13(23):3695. <https://doi.org/10.3390/ani13233695>*

We adapted the sentence to the reviewers suggestion (incl reference) as follows: (L103)

“Although the high degree of female power over males may be a derived trait (Lewis et al 2023), in bonobos the actual underlying mechanisms are still debated.”

use power instead of dominance here

Done

use ranking higher instead of dominating here

Done

use power instead of dominance here

Done

use power instead of dominance here

Done

use power or leverage instead of dominance here

Done

use power or leverage instead of dominance here

Done

use power instead of dominance here

Done

for clarity, replace “as” with “because”

Done

130-7 The authors use the term dominance correctly within the power framework here. They may want to use the general term “power” for consistency, but it fits the framework either way.

We changed it to power for consistency

138,144 Here and throughout, inconsistency in whether numbers are spelled out or not

We decided to spell out numbers below 10 (except in technical contexts and with ages), unless they are associated with higher numbers as in the dataset labels (e.g. 30-years-6-communities dataset), and revised the manuscript accordingly.

Methods & Results: averages should have standard deviations or standard errors and ranges should be associated with medians. Decimal places are inconsistent and not biologically meaningful.

We present in the revised version median and range to account for this valid point by the reviewer (eg L502ff)

Line 161 community size was 16.67 individuals – what is 1/100th of an individual?

Converted in the revised version to medians and therefore no longer decimal places

170-2 include the variation in these numbers

Added median and range in the revised version of the manuscript

use power instead of dominance here

Done

use power instead of dominance here

Done

use outranked instead of dominated here

Done

use power instead of dominance here

Done

use power instead of dominance here

Done

186-7 How were these behavioral data collected across sites? Focal? All occurrences for all individuals? How operationalize “conflict” “coalition” etc.

We added the relevant information to the revised version of the manuscript(L509ff):

“At all sites, all occurrences of contact and non-contact aggressions were recorded during follows of focal parties. We refer to conflicts as instances of dyadic agonistic interactions including an aggressor and a victim. Coalitionary aggression refers to instances of agonistic interactions that

include more than a single aggressor. Female coalitionary aggression in bonobos often incites participation and free-riding of additional individuals 46, sometimes even males (28% of the observed female coalitions in this study). We therefore scored each coalition involving more than one female in the role of primary aggressor as female coalition.

use outranked instead of dominated here

Done

use power instead of dominance here

Done

use power instead of dominance here

Done

Use "because" instead of "as"

Done

use power instead of dominance here

Done

delete "dominance"

Done; replaced with interaction

change "are" to "were"

Done

aggression is usually singular and plural

deleted "s"

does this mean that sometimes the female coalitions include males as well? Mixed-sex?

See rational outlined above

227 use power instead of dominance here

Done

Use "because" instead of "as"

Done

use power instead of dominance here

Done

Change to "where males showed a submissive reaction to" or something similar

We changed it accordingly. The only place where we maintain the word win is in the labelling of the models, as this makes it easier for the reader.

Change to "conflicts in which the female was submissive to the male and the number of conflicts where the male was submissive to the female"

Done

262-3 Change to "males submitted to females in 1099 conflicts and females submitted to males in 687 conflicts"

Done

use power instead of dominance here

Done

MF as a code for maximally tumescent females is confusing for readers who typically think of MF as male/female or multifemale, maybe MTF?

Changed in revised version of the manuscript as suggested by the reviewer

use power instead of dominance here

Done

what is observation level?

A unique number for each observation year in each community. added to the method section of the revised version of the manuscript (L588ff)

use power instead of dominance here

Done

rephrase “won” to refer to submission

Done

only 2 observations...??? Of what? Also, what can really be said about only a handful of observation over multiple years?

We revised the language to specify that this refers to observation years.

re-run? Re-ran?

yes

318-21 This sentence is confusing.

We changed it to (L662ff): “ Males might only consider female coalitions as threats that are directed at males. We wanted to rule out that different quantifications of the propensity of females to form coalitions affected the results of this study.

use power instead of dominance here; rerun? Reran? Re-ran?

Done

Bonobo females often had power over males

Done

329-30 use power instead of dominance here; change to “...intersexual conflicts in which males submitted...dataset, male bonobos submitted to females on average in 61%...”

Done

Change to “percentage of intersexual conflicts in which males submitted to females with ...”; this is HUGE variation! Was the submission to just 1 female and not a coalition of females? This result needs to be better addressed explicitly.

To address the concerns of the reviewer we added the number of intersexual dyads that exhibited a conflict per community (L166ff). This number is on average 16 and has a minimum of 4, allowing us to exclude that conflicts within a single dyad drive the results that we see.

use power instead of dominance here

Done

delete “dominance”

Done

Change to “...measures of female power over males, the proportion of conflicts in which males submitted to females...”

Done, second part see above

347 use power instead of dominance here

Done

Change to “conflicts where females received submission from males within a community...”

Done

use power instead of dominance here

Done

Rephrase “won” to refer to submission

Done

use power instead of dominance here

Done

Figure 1 y-axis: change to “proportion of intersexual conflicts in which males submit”

Done, changed it to “in which females received submission from males” to keep female perspective

delete “degree of female dominance over males measured as the”

Done

Change “won” to “in which males submit to females”

Done

use power instead of dominance here

Done

Rephrase “won”

Done

use power instead of dominance here

Done

Rephrase “won”

Done

use power instead of dominance here

Done

use outranked instead of dominated here

Done

weak evidence? The result is not significant!

Revised, we originally adopted the language of evidence, that are outlined in Muff, S., Nilsen, E. B., O’Hara, R. B. & Nater, C. R. Rewriting results sections in the 766 language of evidence. Trends in Ecology & Evolution 37, 203–210 (2022).

little evidence? The result is not significant!

Revised, we originally adopted the language of evidence, that are outlined in Muff, S., Nilsen, E. B., O’Hara, R. B. & Nater, C. R. Rewriting results sections in the 766 language of evidence. Trends in Ecology & Evolution 37, 203–210 (2022).

use power instead of dominance here

Done

Rephrase “won”

Done

use power instead of dominance here; there are multiple decimals in these values.

Done and corrected

use outranked instead of dominated here

Done

use power instead of dominance here

Done

moderate evidence based on $p=0.036$ and weak evidence based on $p=0.083$? The phrasing throughout the Results with weak or little or moderate evidence based on the p-value is concerning. Maybe plot effect size and talk about strong, moderate, or weak effects? If the authors want to talk about the direction of an effect was consistent with expectations even if it wasn’t significant, I’m more ok with that than the current little/weak/moderate/strong language.

We appreciate the reviewer’s concerns and while our initial approach was based on clearly defined principles known as the “language of evidence,” as outlined in Muff, S., Nilsen, E. B., O’Hara, R. B., & Nater, C. R. (2022). “Rewriting results sections in the language of evidence.” Trends in Ecology & Evolution, 37, 203–210, we now reformulated the presentation of our results, using the term significant and not significant and the direction/strength of the effects (L216ff).

why spell out and not use the FFC abbreviation?

We spell out the terms first time we use them in the result section and subsequently use the abbreviation in the revised version of the manuscript

strong evidence based on p-value? See above.

See above

424 use power instead of dominance here

Done

calling a nonsignificant result as weak evidence is problematic

see above

430 use power instead of dominance here

Done

Figure 2 y-axis: y-axis: change to “proportion of intersexual conflicts in which males submit” and “David’s Scores”

Done, here in the captions we used the female perspective now...proportion of conflicts in which females received submission from males

use power instead of dominance here

Done

adjust text to fit the new y-axis title Table 2 adjust text to fit new language

Done

use power instead of dominance here

Done

456-7 adjust text to fit new language

Done

use power instead of dominance here

Done

use power instead of dominance here

Done

Change to “ranking higher than males”

Done

Use “because” instead of “as”

Done

use power instead of dominance here

Done

“behavioral tool” is an odd word choice. The power framework would suggest that it is not a behavioral tool but just a different base of power. I don’t think the authors need to apply the framework terminology here unless they want to, but I would phrase differently. Strategy?

Yes, we like the word strategy better! (L295) and abstract (L29)

use power instead of dominance here

Done

which indices? There was the FDI-DS, what were the other indices? Or is this across communities? Make sure wording is consistent here “dominance indices”; How did the Wamba study define “won”?

Thank you for your insightful comment. It prompted us to revisit the data from the original citation to ensure that the aggressive interactions aligned precisely with our definitions. Upon re-evaluation, we discovered that when considering only aggressive interactions that elicit submissive behavior, females won 38% of the time. While this figure still falls within the range of our initial

measures, we have adjusted this number in the revised version of the manuscript accordingly(L419ff).

481-2 awkward wording; change to “a male had the highest rank...”; was the mother present?

Changed wording accordingly

How often were males with mothers in this study? Is being with the mother a “behavioral tool” for a male?

We added this information to the revised version of the manuscript (L310ff):

“In three communities—Ekalakala, Bompusa West, and possibly Bompusa East—the mothers of the highest-ranking males were present. In contrast, in the remaining two communities, Kokoalongo and Fekako, none of the adult males had a living co-residing mother.”

use greater power instead of higher dominance here

Done

use power instead of dominance here

Done

relevance or consequence?

Both, changed it to consequence to make it easier

Change to “ ...a given female’s individual rank...”

Done

ownership?; change to “intersexual rank”; the sentence is confusing

Done

delete dominance

Done

already in the their process of reaching adulthood? Odd wording

changed to “during adolescence”

so why is this a hypothesis? What about other “female dominant” species and coercive mating?

We reformulated this sentence and added the lemur perspective later in the discussion (L396ff)

delete dominance

Done

leadership is a completely different phenomenon and seems to come out of nowhere. Females in some “male dominant” species lead group movement.

Removed this part

use power instead of dominance here
typo

corrected

512-3 use power instead of dominance here; Do we know if there are winner/loser effects in these species? Do we know if male-male conflicts increase in frequency as male numbers increase? Key assumptions are not addressed.

Based on the comment by the reviewer we added this information into the discussion of the revised version of the manuscript (L334ff):” In line with further predictions of the model, studies in vervet monkey and capuchin monkey confirmed that a higher proportion of males within groups is not only linked to increased conflict rates among males, but also a greater proportion of female victories over males compared to all other adult pairings 11.”

The key assumptions should be addressed prior to this point and it should be discussed in the Intro about where bonobos fit and where they don't. Bonobos do live in permanent groups, just not cohesive groups. Is the assumption of DomWorld permanent groups or cohesive groups?

In response to this comment, we have 1) relocated the key assumptions of the DomWorld model—specifically those that pertain to bonobos—into the introduction section (L109ff); 2) revised the discussion section addressing the influence of fission-fusion dynamics on the potential impact of grouping patterns on self-organization, particularly in relation to winner-loser effects (L340ff). The reviewer rightly identified our error in initially describing these groups as "not permanent" rather than "less cohesive."

What is the relevant level for assessing sex ratio? How relate to the findings about sex ratios in reference number 24?

We appreciate the reviewer for suggesting an alternative scenario regarding how the group's sex ratio can influence female leverage and, consequently, affect the degree of female power. We now include this scenario in the introduction of the revised version of the manuscript (L73ff): "An alternative explanation to the link between sex-ratio and female power within a group might be that a decreasing female biased sex-ratio changes the supply and demand for mating opportunities and reduces the leverage of females²⁵." Furthermore, we discuss our non-finding of a link between sex-ratio and female power in the light of this alternative hypothesis (L379ff): "In bonobos, factors such as year-round reproduction, confused ovulation, and promiscuous mating may change the dynamics of mate competition, and the sex ratio or even the ratio of males to maximally tumescent females becomes less indicative of male mating opportunities."

buildup? What does that mean?; use power instead of dominance here; and why is this statement true?

We changed the wording to emergence

use power instead of dominance here

Done

more than ?; more active versus a passive role?

We see that this formulation is unfortunate. We changed it in the revised version of the manuscript to express the idea that female coalition formation provides a behavioral strategy for females to change their power within groups (L354).

use power instead of dominance here

Done

use power instead of dominance here; needs a citation; this phrasing places the majority of the agency with males and females appear to be passive recipients without the ability to have any influence on outcomes. Do the authors mean to imply that females do not have much agency?

Done, added a citation by Furuichi (2011) that discusses such an evolutionary scenario (L366), we disagree with the interpretation of the reviewer that females are passive agents, as the basis of all changes are changes in female sexual signaling in this species. We modified the sentence in the revised version of the manuscript to clarify this aspect.

change to "a species for which females sexual swellings..."

Done

why do the authors all of a sudden decide to use the term "power" here?

we now applied the concept to the whole manuscript

typo

deleted 'in'

see previous comment about status and whether it needs to have the descriptor "dominance" in front of it.

removed dominance

but lemurs still exhibit forced copulation despite "female power". It is not a zero-sum game with either males or females having complete autonomy. I think this discussion needs a little more nuance and discussion about the context of power.

We added here (L399ff) " Finally, it is important to note that rarely one sex has complete autonomy over reproduction and it is therefore not surprising to observe rare cases of forced copulations⁸⁷ and infanticide even in lemur species with high female power⁸⁸."

Use "because" instead of "as"

Done

also?; use "because" instead of "as"

Done

"This makes it" What is "this" and what is "it"?

We changed the wording

Use "because" instead of "as"

Done

contraction

Thank you for spotting this. We added the information that observer's intuition about female tendencies to form coalitions are tied to specific observation years (not communities in general), which makes it no longer contradictory.

delete dominance

Done

use power instead of dominance here

Done

delete dominance

Done

use power instead of dominance here

Done

use power instead of dominance here

Done

use power instead of dominance here

Done

This needs a noun after it

added "finding"

use power instead of dominance here

Done

use power instead of dominance here

Done

Supplemental material

Table S1 is confusing:

Headings:

-what is agi?

"Aggression", changed in revised version of the manuscript

-for observation days, what is 64 days for 20 individuals? How many hours of observations with each individual or each possible dyad? And how are there dyadic data without any days of observation?

We changed/explain better the column names in the revised version of the manuscript (L933ff):

1) Observation days are the number of days that bonobos were followed in a given year and relevant for the measure of number of female coalitions (observation effort). This information is now added to the table as N for the proportion of days with more than 1 MTF)

2) Dyads are the number of dyads in which agonistic interactions were observed. We show this number to make sure that it is not one given dyad that basically defines the degree of female power over males. We changed the column name to make this more explicit in the revised version of the manuscript.

-Female coalitions: any information on the response to the coalition?

All with fleeing...added this information to the revised version of the manuscript

-Do the authors mean % or proportion? All of the values are between 0 and 1.

Proportion

842-4 use power instead of dominance here

Done

use outranked instead of dominated here

Done

adjust wording "wins"

Done

adjust wording "won" and "dominated"

Done

Table S2 & S3 adjust wording

Done

RESPONSE TO REFEREES

We want to thank both reviewers for the time and effort they invested into our manuscript

Find below the detailed replies to each inquiry in bold italics

REVIEWERS' COMMENTS:

Reviewer #1 (Remarks to the Author):

No further comments.

Reviewer #2 (Remarks to the Author):

The authors have done a pretty good job revising the manuscript. I have just a few comments, which are mostly copyedits rather than content. The manuscript needs some cleaning up: There are some missing periods at the end of sentences, extra periods at the end of sentences, missing hyphens in hyphenated words, missing spaces at the ends of sentences, etc.

1. The authors said they addressed the sentences with weak structure but they still have numerous sentences, perhaps that were added in the revision and not caught?

Line 26, 63, 480 "This" is a descriptor and must be followed by a noun.

Done

Lines 124, 297, 380, 381, 391, 413, 451, 452 "It" is a pronoun and must refer to a noun.

Rephrase for improved clarity.

done

Lines 186, 206, 212, 237, 247 "there was..." needs to be revised for improved clarity.

done

Lines 287, 288, 296, 322 "there are..." needs to be revised for improved clarity.

done

Line 300, 304, 308, 354, 397 "there ..."

done

2. Verb tense.

Line 81 "rise" should not be in the present tense

done

Line 91 change "is" to "was"

done

Line 111 change "predict" to "predicted"

done

Line 127 change “expect” to “expected”

done

Line 135 change “predict” to “predicted”

done

Line 184 change “is” to “was”

done

3. Line 86, is it true that female coalition hypothesis is the least discussed? Are there any citations or something that can back up this assertion?

No, this comes from a literature search I made in the framework of this study

4. Line 113, “been [proposed] based on...” is this what the authors are saying? It seems the word proposed was missing.

Correct, done

5. Line 174, 1 sentence does not equal a paragraph.

No longer separate paragraph

6. Line 332-5, this sentence is awkward.

Modified to:” Finally, a key distinction between the previously studied species that align with the predictions of the self-organization hypothesis and bonobos could be that the influence of female coalitions against males overshadows the significance of the male-to-female ratio. »

7. Line 390, typo: replace “where” with “were”

Done

8. Line 409, no contractions in formal writing

Done

9. Line 494, do you mean “elicit” (to get) or “solicit” (to request)?

Elicit..changed accordingly

10. Line 913, the figure caption should mention the A and B panels in the figure.

they actually do